# 120,000 year record of sea ice in the North Atlantic ?

Niccolò Maffezzoli[1,2], Paul Vallelonga[2], Ross Edwards[3,4], Alfonso Saiz-Lopez[5], Clara Turetta[1,6], Helle Astrid Kjær[2], Carlo Barbante[1,6], Bo Vinther[2], and Andrea Spolaor[1,6]

[1]Institute for Polar Sciences, ISP-CNR, Via Torino 155, 30172 Venice Mestre, Italy
[2]Centre for Ice and Climate, Niels Bohr Institute, University of Copenhagen, Juliane Maries Vej 30, Copenhagen Ø 2100, Denmark
[3]Physics and Astronomy, Curtin University of Technology, Kent St, Bentley, WA 6102, Perth, Australia
[4]Department of Civil and Environmental Engineering, UW-Madison, Madison, WI 53706, USA
[5]Department of Atmospheric Chemistry and Climate, Institute of Physical Chemistry Rocasolano, CSIC, Madrid, Spain
[6]Ca' Foscari University of Venice, Department of Environmental Sciences, Informatics and Statistics, Via Torino 155, 30170 Venice Mestre, Italy

Correspondence: N. Maffezzoli (niccolo.maffezzoli@unive.it)

**Abstract.** Although it has been demonstrated that the speed and magnitude of recent Arctic sea ice decline is unprecedented for the past 1,450 years, few records are available to provide a paleoclimate context for Arctic sea ice extent. Bromine enrichment in ice cores has been suggested to indicate the extent of newly formed sea ice areas. Despite the similarities among sea ice indicators and ice core bromine enrichment records, uncertainties still exist regarding the quantitative linkages between bromine reactive chemistry and the first year sea ice surface. Here we present a 120,000 year record of bromine enrichment from the RECAP ice core, coastal East Greenland, and interpret it as a record of first-year sea ice. We compare it to existing sea ice records from marine cores and tentatively reconstruct past sea ice conditions in the North Atlantic as far north as the Fram Strait (50-85 °N). We find that during the last deglaciation, the transition from multi-year sea ice to first-year sea ice started at ∼17.5 kyr, synchronous with sea ice reductions observed in the eastern Nordic seas and with the increase of North Atlantic ocean temperature. First-year sea ice reached its maximum at 12.4-11.8 kyr during the Younger Dryas, after which open-water conditions started to dominate, as supported by sea ice records from the eastern Nordic seas and the North Icelandic shelf. Our results show that over the last 120,000 years, multi-year sea ice extent was greatest during Marine Isotope Stage (MIS) 2 and possibly during MIS 4, with more extended first-year sea ice during MIS 3 and MIS 5. Sea ice extent during the Holocene (MIS 1) has been less than at any time in the last 120,000 years.

## 1 Introduction: $Br_{enr}$ as a potential indicator for past sea ice conditions

The connection between Arctic sea ice and bromine was first identified through an anticorrelation between springtime ground level ozone ($O_3$) and filterable bromine air concentrations (Barrie et al., 1988). Large bromine oxide ($BrO$) column enhancements and simultaneous tropospheric ozone depletion were later found in Antarctica (Kreher et al., 1997). Satellite observations reveal geographically-widespread 'bromine explosions', the sudden increase of atmospheric bromine concentrations during springtime occurring in both polar regions (Chance, 1998; Richter et al., 1998; Wagner and Platt, 1998). The mechanism proceeds via springtime photochemical heterogeneous reactions that lead to the activation of bromide, followed by the release

and exponential increase of gas-phase bromine species in the polar troposphere (Vogt et al., 1996). Several saline substrates on fresh sea ice surfaces (hereafter referred as first-year sea ice, FYSI) were suggested as reservoirs of reactive sea salt aerosols (SSA) capable of sustaining bromine recycling (Abbatt et al., 2012; Saiz-Lopez and von Glasow, 2012). To date, both model studies (Yang et al., 2008, 2010) and experimental evidence (Pratt et al., 2013; Zhao et al., 2016; Frey et al., 2019) consider the
deposited snow layer on FYSI (known as salty blowing snow) to be the most efficient substrate for SSA release and bromine activation.

Atmospheric bromine and sodium thus originate from both oceanic and FYSI sea salt aerosol, where their concentration ratio is that of sea water mass: $(Br/Na)_m$ = 0.0062 (the subscript indicates 'marine', Millero et al. (2008)). $Br_{enr}$ values in ice core records, i.e. the bromine-to-sodium mass ratios beyond the sea water value (Eq. 1) were introduced by Spolaor
et al. (2013b) as a potential proxy for past FYSI conditions within the ocean region influencing the ice core location. The basic assumption behind this idea is that bromine recycling on FYSI surfaces would increase the bromine-to-sodium mass ratio beyond the sea water value in the atmosphere and at the ice core location, thus isolating effect of FYSI-induced bromine recycling from the sea salt aerosol contribution, the latter being interfered by oceanic emissions. Further studies have provided some evidence on the validity of $Br_{enr}$ as a FYSI indicator (Spolaor et al., 2013a, 2014, 2016a). At present, however, this
interpretation is still challenged by several uncertainties related to the different variables which play a role in the involved chemical and physical processes (Abbatt, 2013). These can be grouped into three categorizes: bromine activation, transport and deposition/postdeposition. They are briefly discussed here.

Pratt et al. (2013) showed that saline snow collected on Arctic tundra or first-year sea ice surfaces can serve as an efficient reservoir for bromine activation. They point out, however, that acidity of snow is a pre-requisite for bromine activation, as well
as internal snow pack air chemistry (e.g. concentration of $^{\cdot}OH$, $NO_3^-/NO_2^-$). A modification in these factors would alter the bromine production efficiency at constant FYSI extent. Aged sea ice (second-year sea ice or older) is possibly contributing to bromine activation due to a non-zero salt content, but the amount of this second order source effect is unknown.

Bromine travels from the source region to the ice core location both in the aerosol and gas phases, while sodium is only found in the aerosol phase. To date, the partitioning of bromine species between the two phases isn't clear. Additionally,
some studies have reported that bromine recycling takes place within the plume even during the transport (Zhao et al., 2016), leading to possible bromine depletion from the aerosol in favor to the gas phase. Overall, the final observed $Br_{enr}$ values in ice samples depend on the relative importance of aerosol and gas-phase bromine. It appears that fine aerosol are enriched in bromine, while coarse particles are depleted (Legrand et al. 2016, Koenig pers. comm.). Additionally, the two phases likely have different atmospheric residence time, thus the final bromine-to-sodium ratio found in the snow might be function of the
transport duration. On this topic, Simpson et al. (2005) experimentally showed that sodium is washed out from the atmosphere faster than bromine, suggesting the role of longer lasting gas-phase bromine. On the other hand, a model run by Spolaor et al. (2013b) showed opposite results. To which extent transport processes can interfere with a source effect is yet to be resolved.

The last set of uncertainties relates to the variables associated to the deposition of bromine and sodium, such as the variability in accumulation rates over long time scales, which would impact the relative importance of the wet and dry contributions of the

two species. Finally, photolytic reemission of bromine from the snowpack could be responsible for bromine loss and decreased $Br_{enr}$ values.

We acknowledge the above-mentioned open questions on the validity of $Br_{enr}$ as a FYSI proxy and clearly point out that further studies are needed to shade light to these uncertainties. We thus proceed with the interpretation of $Br_{enr}$ as a FYSI proxy bearing in mind that other variables could, at present, explain a $Br_{enr}$ record.

Here, we consider the available evidence regarding past sea ice conditions in the North Atlantic, and present a bromine enrichment record from the RECAP ice core, located in coastal East Greenland. Because of its location, RECAP should record the fingerprint of sea ice in the North Atlantic. We compare our the RECAP $Br_{enr}$ record with sea ice reconstructions from five marine sediment cores drilled within the Renland source area: the Fram Strait, the Norwegian Sea and the North Icelandic shelf (Fig. 1).

## 2   Methods

### 2.1   The 2015 RECAP ice core

The RECAP (REnland ice CAP) ice core was retrieved from the Renland ice cap (71° 18' 18" N; 26° 43' 24" W; 2315 m a.s.l.) from May 13th to June 12th, 2015. The ice cap is located on the Renland peninsula and is independent of the main Greenland ice sheet, with fjords to the north and south. A 98 mm diameter ice core was recovered to 584 m (bedrock). The drilling occurred in a dry borehole to 130 m depth and estisol 140 drilling fluid was used for the remaining depth. The record covers the last 120,800 years. The core age model is described in full details in the Supplementary of *Simonsen et al., Nat. Comm., 2019 (accepted)*.

The ice core samples dedicated to mass spectroscopy measurements (n=1205) were collected and automatically decontaminated from a continuous ice core melting system as part of the RECAP Continuous Flow Analysis (CFA) campaign conducted at the University of Copenhagen in Autumn 2015. The ice core was melted at a speed of approx. 3 cm min$^{-1}$ on a gold coated plate copper melter head (Bigler et al., 2011; Kaufmann et al., 2008). Meltwater was collected continuously from the melter head into pre-cleaned polyethylene vials (cleaned with ultrapure water, > 18.2 MΩcm$^{-1}$) at two different depth resolutions. From the ice cap surface to a depth of 535.15 m, samples incorporated ice meltwater spanning depths of 55 cm. From a depth of 535.15 m to the ice cap bedrock the samples integrated 18.3 cm. The time resolution is annual to multicentennial in the Holocene and from centennial to millennial in the glacial section. After collection, the samples were immediately refrozen at -30 °C and kept in the dark until mass spectroscopy analyses to reduce bromine photolysis reactions.

### 2.2   Experimental: determination of Br, Na, Cl, Ca and Mg by mass spectroscopy

The samples were shipped to Ca' Foscari University of Venice (Italy, n=770) and Curtin University of Technology (Perth, Australia, n=435) for determination of bromine (Br), sodium (Na), chlorine (Cl), calcium (Ca) and magnesium (Mg) by Inductively Coupled Plasma - Mass Spectroscopy. Only bromine and sodium were measured in the samples analyzed at the Italian lab,

while all elements were quantified with the Australian setup. The depth and age ranges associated with the Italian samples are: surface-150 m (2015 AD-328 yr b2k); 165-219 m (383-636 yr BP); 234-413 m (727-2857 yr b2k); 441-495 m (3522-5749 yr b2k). 435 samples were measured in Australia. The depth and age ranges associated with the Australian samples are: 150-165 m (328-383 yr b2k); 219-234 m (636-727 yr b2k); 413-441 m (2857-3522 yr b2k); 495-562 m (5749-120788 yr b2k).

University Ca' Foscari of Venice, Italy

Bromine and sodium ($^{79}$Br and $^{23}$Na) were determined by Collision Reaction Cell-Inductively Coupled Plasma-Mass Spectrometry (CRC-ICP-MS, Agilent 7500cx, Agilent, California, USA). The introduction system consisted of an ASX-520 autosampler (CETAC Technologies, Omaha, USA) and Scott spray chamber fitted with a MicroFlow PFA nebulizer. The sample flow was kept at 100 µLmin$^{-1}$. All reagents and standard solutions were prepared with ultrapure water (UPW, 18.2 MΩcm$^{-1}$).

Nitric acid (65% v/v, trace metal grade, Romil, Cambridge, UK) and UPW washes (2 minutes each respectively) were used for background recovery after every sample analysis. The nitric acid washing concentration was lowered to 2%. The experimental routine (standards and calibrations), as well as the overall instrument performance (detection limits and reproducibility) are the same as in Spolaor et al. (2016b).

Curtin University of Technology, Perth, Australia

The analyses were performed by Inductively Coupled Plasma Sector Field Mass Spectroscopy in reverse Nier-Johnson geometry (ICP-SFMS, Element XR, Thermo Fisher, Germany) inside a Class 100 clean room environment at Curtin University TRACE facility (Trace Research Advanced Clean Environment Facility). The ICP-SFMS introduction system consisted of an Elemental Scientific Inc. (ESI, Omaha, USA) syringe-pumped autosampler (Seafast II) with a 1 mL PFA capillary injection loop and using an ultrapure water carrier. A 1 ppb indium internal standard in 5% v/v nitric acid (HNO$_3$, double PFA distilled)

was mixed inline at a flow rate of 25 µLmin$^{-1}$ using a T-split (final flow rate of 400 µLmin$^{-1}$, take-up time 1.5 min). Nebulization occurred in a peltier cooled (2 °C) quartz cyclonic spray chamber (PC3, ESI), fitted with a PFA micro-concentric nebulizer (PFA-ST, ESI). Bromine, sodium and magnesium, chlorine and calcium isotopes ($^{79}$Br, $^{23}$Na, $^{24}$Mg, $^{35}$Cl, $^{44}$Ca) were detected in medium resolution (10% valley resolution of 4000 amu) and normalized to $^{115}$In. Memory effects were reduced by rinsing the system between samples with high purity HNO$_3$ (3%) and UPW. One procedural blank and one quality

controlled standard (QC) were analyzed every 5 samples to monitor the system stability. The detection limits, calculated as 3$\sigma$ of the blank values (n=80) were: 0.18 ppb (Br); 1.1 (Na); 0.2 ppb (Mg); 1.6 ppb (Ca) and 4.6 ppb (Cl). The majority of the sample concentrations were above the detection limits for all elements (>97%). The relative standard deviations of the control standard samples (n=82) concentrations were monitored over >100 hours and were 9% (Br); 4% (Na); 3% (Mg); 4% (Ca) and 13% (Cl). Calibration standards were prepared by sequential dilution (7 standards) of NIST traceable commercial

standards (High-Purity Standards, (Charleston, USA)). All materials used for the analytical preparations were systematically cleaned with UPW (18.2>MΩcm$^{-1}$) and double PFA distilled ultrapure HNO$_3$ (3%, prepared from IQ grade HNO$_3$, Choice Analytical Pty Ltd, Australia) throughout.

A laboratory intercomparison between sodium and bromine measurements was performed on a common set of samples (n=140) to investigate differences between the analytical techniques and laboratories, as described in Vallelonga et al. 2017.

The correlations and the gradients between the measured concentrations in the two setups are $\rho(\text{Na}_{IT}\text{-Na}_{AUS})=0.99$ (n=140; p<0.01), $m_{Na}= 1.08\pm0.01$ (1$\sigma$) for sodium and $\rho(\text{Br}_{IT}\text{-Br}_{AUS})=0.93$ (n=140; p<0.01), $m_{Br}=1.08\pm0.02$ (1$\sigma$) for bromine.

## 2.3 Atmospheric reanalysis: the source region of bromine and sodium for the RECAP ice core

To estimate the source of bromine and sodium deposited at Renland, daily back trajectories were calculated from 2000 AD to 2016 AD with HYSPLIT4 (Stein et al., 2015; Draxler et al., 1999; Draxler and Hess, 1998, 1997), using publicly available NCEP/NCAR Global Reanalysis meteorological data (1948-present), with a 2.5° resolution in both latitude and longitude (Kalnay et al., 1996). The back trajectories were started daily on an hourly basis at 500 m above the Renland elevation (71.305 °N, 26.723 °W, 2315 m a.s.l.) for the 17 year time span. The trajectory time was set to be 72 hours, representing the average atmospheric lifetime of sea salt aerosol (Lewis and Schwartz, 2004) and likely a lower limit for inorganic gas-phase bromine compounds. To access the potential marine sources of bromine and sodium, only a subset of the all trajectories is considered. Such selection limits the ensemble to only such trajectories that crossed the marine boundary layer (MBL), defined here as the 900 hPa isosurface (corresponding to approximately 1000 m a.s.l.), for at least 10 hours. This pressure value was chosen according to Lewis and Schwartz (2004) and references within. One way to display a map of MBL layer crossings is the Residence Time Analysis, a spatial distribution of trajectory endpoints (Ashbaugh et al., 1985). This map indicates that 75% of the signal originates from the North Atlantic Ocean, extending in latitude from 50° N to 85° N (up to the Fram Strait) and in longitude from the western coasts of Norway and the UK to East Greenland (Fig. 1), although the ocean regions closer to Renland are expected to be more significant as per the observed chemical signature. A minor contribution is expected from aerosols and gas-phase bromine originated from coastal waters off West Greenland. The consistency of sea ice reconstructions from the RECAP core and the Nordic Sea sediment cores (Sect. 4.2) suggests that the source area could extend to these regions throughout the last 90 kyr. For the overall interpretation of the RECAP record, the source area is therefore assumed to be the 75% contour region of the Residence Time Analysis endpoint distribution (Fig. 1). Such region is nowadays mostly dominated by open water (OW) conditions, with only minor contributions of FYSI grown in situ and MYSI transported south from the Arctic Ocean alongside the East Greenland coastline via the Transpolar Drift (Fig. 2).

## 3 Calculation of bromine enrichment ($\text{Br}_{enr}$) and $\text{Br}_{enr}$ time series

The bromine enrichment values in the ice samples are calculated from the departure from the sea water abundance, the latter inferred from sodium (hereafter sea-salt sodium, ssNa):

$$\text{Br}_{enr} = \frac{Br/ssNa}{(Br/Na)_m} \tag{1}$$

$$\text{Na} = ssNa + nssNa \tag{2}$$

where Br, ssNa and nssNa are the bromine, sea-salt and non-sea-salt sodium concentrations in the ice samples respectively, and $(Br/Na)_m = 0.0062$ is the bromine-to-sodium mass ratio in sea water (Millero et al., 2008), assumed constant in time

and space. Since sodium concentrations in ice cores can be interfered by terrestrial inputs (nssNa, from sodium oxide $Na_2O$), which can generally contribute up to $\approx$10-50% during glacial arid periods (i.e. stadials), nssNa and ssNa have to be evaluated to calculate $Br_{enr}$ (Eq. 1-2). We estimate the ssNa concentrations by using three methods. They make use of chlorine, magnesium and calcium concentrations.

The decrease of the Cl/Na mass ratio from the sea water reference (1.8 from Millero et al. 2008) can be related to extra sodium inputs of terrestrial origin. Thus, the ssNa can be calculated from chlorine if the measured chlorine-to-sodium ratio becomes less than 1.8, while zero nssNa is assumed otherwise (Eqs. 3-4):

$$ssNa = \frac{Cl}{1.8} \longleftrightarrow \frac{Cl}{Na} < 1.8 \tag{3}$$

$$ssNa = Na \longleftrightarrow \frac{Cl}{Na} \geq 1.8 \tag{4}$$

Unfortunately, sea salt aerosols are affected by acid-scavenged ($HNO_3$ and $H_2SO_4$) dechlorination processes (e.g. $2NaCl + H_2SO_4 \rightarrow 2HCl + Na_2SO_4$), in which chlorine is removed as gas-phase HCl. These processes would increase the Cl/Na ratios in ice beyond the sea water reference (Legrand and Delmas, 1988), especially in warmer climatic conditions, since HCl is believed to have a longer atmospheric residence time than SSA. In fact, RECAP Holocene Cl/Na ratios ($\approx$ 3) suggest that dechlorination processes do occur and chlorine is also deposited as HCl. Additionally, this result shows that HCl loss from the

snowpack after deposition is limited, opposite to what is observed in Antarctica (Dome C) (Röthlisberger et al., 2003). The huge difference between Dome C and Renland accumulation rates likely explain such differences. In glacial ice, and especially during MIS2 and MIS4, the effect of dust neutralization of nitric and sulfuric acids would reduce chlorine loss both after deposition and during transport, therefore making the above correction more trustworthy (Wolff et al., 1995). However, the relative contributions of dechlorination processes and dust loading as per the observed Cl/Na ratios in the ice are difficult to

quantify. A correction based on Cl/Na ratios was used by Hansson (1994) to calculate the ssNa and nssNa concentrations in the 1988-drilled Renland core. In particular, they found that nssNa accounts for respectively 17% and 24% of the total sodium during MIS2 and MIS4. If the same correction is applied to our measurements, we find values of 15% and 20-30% during MIS2 and MIS4, close to their findings.

    A better alternative to calculate ssNa and nssNa would be to use an element of terrestrial origin. Magnesium contains both

a marine and a terrestrial signature. By using reference mass ratios of Na/Mg in sea water and in the Earth upper continental crust, we are able to calculate the marine sodium contribution (Eq. 7). Often in ice core studies, the sea-salt and non-sea-salt contributions are calculated using global-mean reference values of chemical composition of terrestrial elements (see for example Rudnick and Gao 2003, Table 1). However, the spatial variability of the dust mineralogy can heavily impact the results. In our case, the terrestrial value $(Na/Mg)_t$ to be used in the calculation (the subscript 't' refers to 'terrestrial') would

require knowledge of the geochemical composition of the dust deposited at Renland. In remote Greenlandic ice cores, the provenance of aeolian mineral dust is believed to be the deserts in North East Asia, as inferred from Sr, Nd and Pb isotopic ratios (Biscaye et al., 1997). We therefore carried out a compilation of present day Na/Mg mass ratios of dust from North East Asian deserts, assuming that such an Asian dust source remained the dominant one over the last 120 kyr. For such analysis, the reader is referred to the Appendix (A), that considers n=6 studies of dust composition (Na, Mg and Ca) from the Gobi and

desert regions of Mongolian and northern China deserts. This analysis (Appendix A) shows that, on average, dust from these regions exhibits the following ratios:

$$(\mathrm{Na/Mg})_t = 1.23 \tag{5}$$

$$(\mathrm{Na/Ca})_t = 0.38 \tag{6}$$

It is worth noting that these values differ substantially from global average values (Rudnick and Gao, 2003), especially for Na/Ca.

By using the sodium-to-magnesium ratios in asian dust $(\mathrm{Na/Mg})_t = 1.23$ (Eq. 5) and in sea water $(\mathrm{Mg/Na})_m = 0.12$ (Millero et al., 2008) (the subscript 'm' refers to 'marine'), we are able to calculate ssNa and ssMg (Eqs 7-8):

$$\mathrm{ssNa} = \frac{Na - (\mathrm{Na/Mg})_t \cdot Mg}{1 - (\mathrm{Mg/Na})_m \cdot (\mathrm{Na/Mg})_t} \tag{7}$$

$$\mathrm{ssMg} = \frac{(\mathrm{Mg/Na})_m \cdot Na - (\mathrm{Mg/Na})_m \cdot (\mathrm{Na/Mg})_t \cdot Mg}{1 - (\mathrm{Mg/Na})_m \cdot (\mathrm{Na/Mg})_t} \tag{8}$$

In the Holocene, on average ssNa≈0.99Na, while ssMg≈0.50-0.90Mg. During MIS2, ssNa≈0.6–0.7Na and ssMg≈0.20–0.30Mg, while during MIS4 ssNa≈0.50–0.60Na and ssMg≈0.10–0.20Mg. In those Holocene samples where magnesium was not measured, we assume Na = ssNa, making on average a 1% error.

   The same procedure using calcium has been used to calculate ssNa from Antarctic ice cores (Röthlisberger et al., 2002), by

solving the equations similar to Eq. 7 and Eq. 8 with Mg replaced by Ca. In the RECAP record, such correction (with asian dust $(\mathrm{Na/Ca})_t$=0.38 and $(\mathrm{Ca/Na})_m = 0.038$ (Millero et al., 2008) unrealistically predicts ssNa $\approx 0$ throughout the glacial period. Small changes in the $(\mathrm{Na/Ca})_t$ value do not significantly change this result. It appears that higher-than expected calcium is deposited at Renland, and therefore a lower $(\mathrm{Na/Ca})_t$ is to be used. Excess of calcium during the glacial period, has been reported in Greenlandic ice cores, in the form of calcite ($\mathrm{CaCO_3}$), dolomite ($\mathrm{CaMg(CO_3)_2}$) and gypsum ($\mathrm{CaSO_4 \cdot 2H_2O}$)

(Mayewski et al., 1994; Maggi, 1997; De Angelis et al., 1997). In their investigation of the GRIP core Legrand and Mayewski 1997 find larger increase in sufate ($\mathrm{SO_4^{2-}}$) in Greenland compared to Antarctica during glacial times. Such large enhancements of the sulfate level are well correlated with calcium increases, but not with MSA, suggesting that sulfate levels are related to nonbiogenic sulfur sources (gypsum emissions from deserts, for instance) (Legrand and Mayewski, 1997). Different calcium sources were likely active during the glacial as compared to warmer periods, but it is unclear whether these were new active

continental sources or continental shelfs which became exposed due to a lowered sea level. Similarly to what has been observed in other Greenland ice cores, extra calcium is also found in the RECAP core during the glacial, as illustrated from the scatter plot with magnesium (Fig. 3). From the calcium-magnesium relation, we estimate that during the coldest and most arid glacial times $(\mathrm{Ca/Mg})_t = 23 \pm 1$ (blue fit in Fig. 3), while in other periods we assume that the composition remained the same as the modern Asian dust composition (Appendix A):

$$\left(\frac{Ca}{Mg}\right)_{t,modern} = 3.7 \tag{9}$$

This hypothesis is substantiated by the value found for the lower envelope for the Holocene, interstadial late MIS5 samples (green points in Fig. 3) as well as for composition of the bottom 22 meters of the core (562–584 m, red points in Fig. 3):

$(Ca/Mg)_t = 3\pm1$ (red fit). We note that these bottom measurements do not appear in any time series since the core chronology ends at 120 kyr (562 m), although we here suggest that the RECAP bottom 22 meters may date back to the previous interglacial, the Eemian.

Sea-salt sodium concentrations (ssNa) can therefore be calculated using calcium, by replacing in Eq. 7 Mg→Ca and by using $(Ca/Na)_m = 0.038$ and $(Na/Ca)_t$ calculated by:

$$\left(\frac{Na}{Ca}\right)_t = \frac{(Na/Mg)_{t,modern}}{(Ca/Mg)_{t,glacialfit}} = \frac{1.23}{23} = 0.054 \tag{10}$$

This latter value is not too far to the value (0.036) empirically found by De Angelis et al. 1997, who calculated ssNa in the GRIP core based on a mixed chlorine-calcium method. The calculation of RECAP nssCa reveals that calcium contains almost a purely crustal signature: $nssCa \gtrsim 0.95Ca$ throughout the record.

The ssNa curves calculated with magnesium and calcium ($ssNa_{Mg}$, $ssNa_{Ca}$) differ by less than $1\sigma$, while appreciable differences between these two curves and the chlorine one ($ssNa_{Cl}$) are only found during MIS2 and MIS4 (Fig. 4b). Since $ssNa_{Mg} \simeq ssNa_{Ca}$, for the following discussion we will only consider the two $Br_{enr}$ series based on the chlorine and magnesium correction: $Br_{enr,Cl}$ and $Br_{enr,Mg}$ (Fig. 4g). Standard error propagation is carried out to yield the final $Br_{enr}$ uncertainties. This analysis along with past studies demonstrate that the sea-salt and non-sea-salt calculations of elemental concentrations depend on the chemical composition of the dust that is deposited at the ice core site. The dust composition has a spatial variability which should be considered, while the use of tabulated reference values should be discouraged.

From the calculated RECAP $Br_{enr}$ curves, we now investigate past sea ice conditions in the 50-85 °N North Atlantic Ocean, based on the aforementioned hypotheses on the $Br_{enr}$ use as an indicator of first-year sea ice conditions. Because of the RECAP location, the record is sensitive to ocean processes and sea ice dynamics (Cuevas et al., 2018). We compare our sea ice record with $PIP_{25}$ results from three marine sediment cores drilled within the Renland source area: the Fram Strait, the Norwegian Sea and the North Icelandic shelf. The $PIP_{25}$ index is a semi-quantitative indicator of the local sea ice condition at the marine core location. It is calculated by coupling the sediment concentration of $IP_{25}$, a biomarker produced by diatoms living in seasonal sea ice, with an open water phytoplankton biomarker (brassicasterol or dinosterol, hence $P_BIP_{25}$ or $P_DIP_{25}$ ). Briefly, the $PIP_{25}$ index is a dimensionless number varying from 0 to 1: $PIP_{25} \approx 1$ indicates perennial sea ice cover; $PIP_{25} \approx 0$ indicates open water conditions, while intermediate $PIP_{25}$ values reflect seasonal sea ice (Müller et al., 2011; Belt and Müller, 2013).

## 4   Results and discussion

The two $Br_{enr}$ time series ($Br_{enr,Cl}$ and $Br_{enr,Mg}$), based on either chlorine or magnesium for the for sea-salt sodium calculation, display values greater than 1 throughout the record, suggesting some FYSI signature throughout the last 120 kyr in the North Atlantic (Fig. 4g). The very low values found in the Holocene (on average $Br_{enr,Hol} = 3.9$, RMS(root mean square) = 1.5) suggest minimum FYSI during the current interglacial. Although the record does not extend to the Eemian period, it shows that at the inception of the last glacial period, ∼120,000 years ago, $Br_{enr}$ values were higher than the Holocene (on average $Br_{enr,120kyr} = 5.4$, RMS = 1.1, n=38), suggesting that more FYSI was present at that time. Since 120 kyr ago the $Br_{enr}$ levels

increased until the Greenland Interstadial 21 (GI-21), $\sim$85-80 kyr ago ($\mathrm{Br_{enr,85-80kyr}} \simeq 8$). Hereafter, they decreased throughout late MIS 5. From late-MIS 5 to MIS 4, rising $\mathrm{Br_{enr}}$ values are observed, but a statistically significant difference between the two $\mathrm{Br_{enr}}$ curves result from the two ssNa corrections, with the $\mathrm{Br_{enr,Mg}}$ series showing higher values than $\mathrm{Br_{enr,Cl}}$. The two series converge again at the beginning of MIS 3, with values around 7$\pm$1. From $\sim$ 50-40 kyr both series start to decrease.

The decrease is more significant for $\mathrm{Br_{enr,Cl}}$. Minimum $\mathrm{Br_{enr}}$ values are reached at the Last Glacial Maximum during MIS 2. The deglaciation reveals a $\mathrm{Br_{enr}}$ increase up to ca. 12,000 years ago, followed by a steady decrease towards the Holocene.

The first Arctic glacial-interglacial investigation of $\mathrm{Br_{enr}}$ was performed on the NEEM ice core, located in Northwest Greenland. At NEEM, the main source regions for sea ice-related processes are believed to the Canadian archipelago, the Baffin Bay and the Hudson Bay (Spolaor et al., 2016b; Schüpbach et al., 2018). The 120,000 year NEEM record (Spolaor

et al., 2016b) showed higher $\mathrm{Br_{enr}}$ values during the Holocene and interstadials, compared to the glacial stadials (Fig. 4f). This was interpreted with greater FYSI conditions existing during such warmer periods, while the lower $\mathrm{Br_{enr}}$ values during colder climate phases would indicate more extended multi-year sea ice (MYSI) coverage in those ocean regions. In accordance with their respective source regions, the $\mathrm{Br_{enr}}$ record from the NEEM ice core is markedly different from the RECAP one (Fig. 4). In the NEEM core, $\mathrm{Br_{enr}}$ was positively correlated to $\delta^{18}O$ throughout the entire climatic record. In contrast, at Renland, such

a consistent correlation is not present, and $\mathrm{Br_{enr}}$ is at times lower during warmer climate periods (e.g. during the Holocene). We suggest that this difference originates from the fact that during warm periods, the relative FYSI-to-OW influence is greater at NEEM than at Renland, the latter being mostly dominated by OW conditions in the North Atlantic. This is supported, at least for present conditions, by a model study that investigated the Arctic spatial variability of the ratio of sea ice to open ocean sodium loadings (Rhodes et al., 2018), where it was found that such ratio at NEEM is $\sim$5 times higher than Renland.

Since the increase in atmospheric bromine is believed to reflect the strength of bromine recycling from FYSI surfaces, low values of $\mathrm{Br_{enr}}$ could indicate either OW or MYSI conditions. Thus, we suggest that $\mathrm{Br_{enr}}$ is a signature, at Renland, of contrasting sea ice states: FYSI/MYSI and FYSI/OW. We also suggest that the former (FYSI/MYSI) occurs during generally colder periods while the latter (FYSI/OW) occurs during generally warmer periods. In the framework of this dual regime behavior, if the ice core time resolution is high enough, an increase of $\mathrm{Br_{enr}}$ values, followed by a maximum and a negative

trend should be a signature of an ocean facing gradual changes from open ocean to multi-year sea ice conditions, or viceversa, in case of monotonically warming or cooling ocean temperatures (Fig. 5). One such regime shift is observed in the $\mathrm{Br_{enr}}$ record during the deglaciation (Fig. 4g). At the Last Glacial Maximum (LGM, $\sim$23 kyr), the low enrichment values ($\mathrm{Br_{enr,Mg,LGM}}$ = 3.3$\pm$0.3; $\mathrm{Br_{enr,Cl,LGM}}$ = 3.2$\pm$0.2) suggest reduced FYSI recycling and therefore extensive MYSI conditions (blue arrow in Fig. 5; OW conditions are considered unlikely). Transitioning from the Last Glacial Maximum into the Holocene, $\mathrm{Br_{enr}}$

increases until $\sim$12 kyr ($\mathrm{Br_{enr,Mg,12kyr}}$ = 8.7$\pm$0.9; $\mathrm{Br_{enr,Cl,12kyr}}$ = 7.5$\pm$0.5), indicating maximum FYSI at this time (Fig. 4 and Fig. 6). Hereafter, a decrease is observed, as $\mathrm{Br_{enr}}$ continues to drop during the Early Holocene and the proxy operates in the FYSI/OW regime (red arrow in Fig. 5).

## 4.1 The last deglaciation and the dual $Br_{enr}$ regimes

We now consider in further detail the last deglaciation, when a number of ocean temperature, salinity, circulation and sea ice changes are observed in the Nordic seas (Fig. 6). Marine-derived local sea ice records from both the Svalbard margin and the Norwegian Sea indicate (Fig. 6e,f) that near-perennial sea ice ($PIP_{25} \approx 0.5\text{-}1$) was present during MIS 2 until $\sim$17 kyr (17.6 kyr recorded in the Svalbard margin), the onset of a major breakup of extensive sea ice cover, during Heinrich Event 1 (18 to 15 kyr). Synchronous to within a few centuries, several modifications relevant to the North Atlantic ocean are observed (Fig. 6), including sea water surface freshening and warming in the polar and subpolar North Atlantic (the 67 °N Dokken and Jansen 1999 record is shown as an example in Fig. 6c) and a near total cessation of the Atlantic Meridional Overturning Circulation (AMOC, Fig. 6d). Generally low to intermediate $PIP_{25}$ values ($PIP_{25} \approx 0\text{-}0.5$) are reported in the Svalbard Margin and in the Norwegian Sea in the $\sim$17-12 kyr period (Fig. 6e,f,g), with a slight increasing trend throughout the Bølling-Allerød (BA) and a broad maximum reached during the Younger Dryas (YD), suggesting that seasonal sea ice conditions were dominating this period. Other studies from marine records in the Nordic Seas records also suggest milder sea ice conditions during the BA and increased sea ice during the YD (Belt et al., 2015; Cabedo-Sanz et al., 2016). In contrast, a record from the northern Icelandic Shelf (Fig. 6e) shows that here the sea ice conditions remained near-perennial from 14.7 to 11.7 kyr ($PIP_{25} \approx 0.5\text{-}1$). The authors (Xiao et al., 2017) suggest that this pattern of more severe sea ice conditions in the north of Iceland is, at least during the BA and the YD, linked to the flow of warmer waters from the North Atlantic Current, influencing sea ice melting in the eastern Nordic Seas, whereas the Icelandic shelf is influenced by colder polar waters from the East Greenland Current and the East Icelandic Current.

The RECAP ice core was resampled at sub-centennial resolution to better constrain the timing of sea ice changes through the deglaciation in the 50-85 °N North Atlantic (Fig. 6b, squares). The $Br_{enr,Mg}$ serie ($Br_{enr,Cl}$ would lead to the same results) would indicate that FYSI started to increase in the North Atlantic, concurrent to a reduction of MYSI, at $\sim$17.5 kyr, synchronous with local $PIP_{25}$ decrease in the Svalbard margin and eastern Nordic Seas and in response to sea surface temperature warming in the North Atlantic. This finding would also suggest that North Atlantic sea ice changes occurred in concert with temperature and circulation changes of the underlying surface waters. We note that this time period also coincides with the initiation of deglacial changes in mean ocean temperature, Antarctic temperatures and atmospheric $CO_2$ concentrations toward interglacial values (Bereiter et al., 2018). North Atlantic FYSI continued to increase throughout the BA (except for one point at 12.7-12.4 kyr at the onset of the YD) until a maximum at 12.4-11.8 kyr during the YD, when a clear $Br_{enr,Mg}$ maximum is observed (Fig. 6b). From the comparison between the marine and ice core results, we infer that, during the 17-12 kyr period, the 50-85 °N-integrated North Atlantic sea ice changed from MYSI to FYSI. Local sea ice was also melting at $\sim$17 kyr in the eastern Nordic Seas, likely influenced by the North Atlantic Current, while, at least from 14.7 to 11.7 kyr, sea ice was still near-perennial at the North Icelandic shelf, possibly due to the influence of cold waters carried by the East Greenland Current. Following its maximum value at 12.4-11.8 kyr, $Br_{enr}$ (i.e. FYSI) started to decrease (Fig. 6b). We suggest that from this point-in-time, the $Br_{enr}$ indicator now shifts to the FYSI/OW regime (Fig. 5), and the North Atlantic basin became largely ice free. A retreating FYSI scenario is also recorded in all 5 marine cores (decreasing $PIP_{25}$ to $\approx 0\text{-}0.4$ during the Early Holocene, Fig. 6e,f,g),

suggesting that open water conditions progressively developed in the whole North Atlantic basin, sustained by increasing heat transport from the North Atlantic Current and a strengthened AMOC since ~11.7 kyr (McManus et al., 2004; Ritz et al., 2013).

Since $Br_{enr}$ is assumed to be an increasing function of FYSI, its decrease would point to either OW or MYSI conditions, following either the FYSI/OW or the FYSI/MYSI regimes (Fig. 5). At any point in time, only one regime is considered to be in place, and we suggest a simple model in which a temperature threshold could be the discriminating variable setting the regime type. Since a change of regime is observed during the deglaciation, with maximum $Br_{enr}$ values (i.e. FYSI) at 11.8-12.4 kyr, we set the threshold to be the mean NGRIP temperature reconstructed for that period: $T_{NGRIP}$=-44.6±0.9 ($2\sigma$) °C (the two lines in Fig. 6a). In every ice sample of the 120,000 year record the regime type (FYSI/MYSI or FYSI/OW, Fig. 5) can thus be determined according to its integrated temperature value with respect to the temperature threshold: FYSI/MYSI for a lower temperature value and FYSI/OW for a higher temperature value.

According to this simple model the deglaciation is characterized by the FYSI/MYSI regime until the onset of the Bølling-Allerød (except few points at which the regime type depends on the chosen threshold value, Sect. 4.2), while the FYSI/OW regime operated from that point forward. We note that there is no similar $Br_{enr}$ maximum at the onset of the Bølling-Allerød as seen in the YD at the point when NGRIP temperature is crossing the same temperature as found in the YD. The NGRIP temperature alone appears therefore not to be able to fully explain the observations. The possible explanation of higher $Br_{enr}$ values during the Younger Dryas compared to the Bølling-Allerød may reside in the higher seasonal temperature variations (Buizert et al., 2014) and freshwater inputs from melting ice sheets in the former period, both promoting the formation of seasonal sea ice. Conversely, the lower $Br_{enr}$ values (hence to greater MYSI in the FYSI/MYSI regime) during the Older Dryas compared to the Bølling-Allerød and the Younger Dryas may be linked to the overall much lower temperatures during this period (Buizert et al., 2014), higher surface water salinity due to less freshwater inputs from melting ice sheets and a generally weaker AMOC.

## 4.2 The 120,000 year $Br_{enr}$ record

We now apply the previously mentioned temperature-based discrimination of the two sea ice regimes to the 120,000 year $Br_{enr,Mg}$ record (Fig. 7). The same analysis performed on the $Br_{enr,Cl}$ curve can be found in the Appendix (Fig. B1). The regime type in each ice sample is represented by the color of the error bars: blue for the FYSI/MYSI regime and red for the FYSI/OW regime. In order to test the sensitivity of the regime output on the threshold value, 4 scenarios are computed, using a ±$1\sigma$ and a ±$2\sigma$ value around the temperature threshold mean value $\overline{T}$ = -44.6 °C (Fig. 7). Except 2 (1) samples at 20 kyr showing a different regime type during the deglaciation (in the $\overline{T}$-2(1)$\sigma$ scenario- top panels in Fig. 7), the regime discrimination mentioned in the following discussion is invariant with respect to the 4 scenarios.

MIS 5 is characterized by increasing $Br_{enr,Mg}$ values in the FYSI/OW regime from 120 to 80 kyr ago (Greenland Interstadial, GI-21). We interpret this trend with increasing FYSI extent in the North Atlantic. From the end of GI-21 the $Br_{enr}$ regime changes to FYSI/MYSI as the NGRIP temperature drops during late-MIS 5 and MIS 4. As compared to the levels reached during GI-21 (8±1), lower $Br_{enr,Mg}$ values are found during GS-21, GS-20 and GS-19 (5±2) and on average during MIS 4 (7±1). We interpret the combined effect of decreased $Br_{enr,Mg}$ values and a FYSI/MYSI regime with an increase of MYSI

extent in the North Atlantic from late-MIS 5 to MIS 4. The $Br_{enr,Cl}$ curve shows even lower values during MIS 4, predicting more extended MYSI than the $Br_{enr,Mg}$ record (Fig. B1). Overall, from 120 kyr ago to MIS 4, the change of $Br_{enr}$ regime (from FYSI/OW to FYSI/MYSI) is opposite to what is observed during the deglaciation (from FYSI/MYSI to FYSI/OW): from a relatively warm climate 120 kyr ago, the cooling trend is characterized by an increasing FYSI extent until the point

(GI-21) in which the FYSI starts to be replaced by MYSI. This maximum $Br_{enr,Mg}$ value at the time of the regime change (GI-21) is similar to the peak $Br_{enr,Mg}$ value reached during the deglaciation: $Br_{enr,Mg}(12\,kyr) \simeq Br_{enr,Mg}(GI\text{-}21) \simeq 8$. The same result holds if one considers the $Br_{enr,Cl}$ serie (Fig. B1). Unlike the longer lasting GI-21, the Greenland Interstadials GI-20 and GI-19.2 display very low $Br_{enr}$ values ($\simeq$ 3-5). A possible explanation for these low values could be found in a fast replacement (at least not captured by the time resolution of the $Br_{enr}$ record) of MYSI by OW conditions. Higher time resolved

measurements would be needed to test this hypothesis.

Moving from MIS 4 to MIS 3, the $Br_{enr,Mg}$ values remain within $1\sigma$ of each other. This suggests similar FYSI extent in the two periods, although the FYSI/MYSI regime operating during MIS 4 could suggest extra MYSI extent during this period, as compared to MIS 3, the latter being generally characterized by a mixture of both $Br_{enr}$ regimes. This hypothesis is supported by the $Br_{enr,Cl}$ series, which shows lower values during MIS 4 than MIS 3 (Fig. B1). A sea ice record in the Norwegian Sea

(Hoff et al., 2016) also indicates more perennial sea ice conditions during MIS 4 than during MIS 3. The time resolution does not allow a deep investigation on DO events during MIS 3, although a low $Br_{enr}$ value in the FYSI/OW regime during the GI-12 ($Br_{enr,Mg}(46.0\text{-}46.6\,kyr)=5.2\pm0.7$) suggests a shift to open water conditions during the interstadial, similarly to GI-19.2 and GI-20. Possibly, similar sea ice dynamics were in play during these DO events. Increased time resolution is however needed to better characterize the DO events.

MIS 2 is characterized by a FYSI/MYSI regime and progressively decreasing values of both $Br_{enr,Mg}$ and $Br_{enr,Cl}$ series (Fig. 7 and Fig. B1), reaching their respective minima at $\sim$23 kyr ago, during the Last Glacial Maximum: $Br_{enr,Mg}(LGM)=3.3\pm0.3$; $Br_{enr,Cl}(LGM)=3.2\pm0.2$. We interpret this negative trend with progressively increasing MYSI conditions in the whole North Atlantic, that reached a maximum during the LGM. Maximum $PIP_{25}$ values are also reported in the Norwegian Sea at this time, indicating perennial sea ice conditions at this time.

The deglaciation is characterized by increasing FYSI conditions until the mid-Younger Dryas (12.4-11.8 kyr ago), followed by a return to the FYSI/OW regime and decreasing $Br_{enr}$ values as open waters progressively replaced sea ice in the Holocene. A clear sea ice decline is also reported in the Norwegian Sea following the LGM towards the current interglacial (Fig. 6 and Fig. 7).

## 5 Conclusions and outlook

We present a 120,000 year record of bromine enrichment ($Br_{enr}$) from the RECAP ice core, and interpret it in terms of first-year sea ice (FYSI) variability in the 50-85 °N-integrated North Atlantic ocean. The record suggests that, during the last deglaciation, sea ice started to transform from multi-year sea ice to first-year sea ice $\sim$17.5 kyr ago, probably triggered by increasing surface ocean water temperatures. Increasing first-year sea ice conditions are observed throughout the Older Dryas,

the Bølling-Allerød and the Younger Dryas. During these periods, availability of freshwater from melting ice sheets, seasonal temperature variations and AMOC strength likely played a role in driving the sea ice changes. The maximum first-year sea ice signature is found 12.4-11.8 kyr ago, during the Younger Dryas, whereupon open ocean was the dominant condition during the Holocene (MIS 1). Although the RECAP sea ice record does not extend back to the warmest period of MIS 5, it does

show that sea ice extent during the Holocene is lower than at any time in the past 120,000 years. Minimum first-year sea ice, likely associated with maximum multi-year sea ice conditions existed during MIS 2 and possibly during MIS 4. Compared to MIS 2, greater first-year sea ice extent existed during MIS 3 and MIS 5. Increased time resolution is needed to fully resolve Dansgaard-Oeschger oscillations. However, the data indicate that during GI-12, GI-19.2 and GI-20, large extent of first-year sea ice could have been replaced by open ocean, while large first-year sea ice areas existed during GI-21.

These analyses and conclusions rely on a number of hypotheses and assumptions mainly concerning the validity of $Br_{enr}$ as a proxy for first-year sea ice. Efforts are still needed in the direction of validating these assumption and investigating the limitations and processes that affect the $Br_{enr}$ signature in present deposition and in old ice core records.

*Data availability.* The RECAP ice core data will be made available on NOAA paleoclimate and PANGAEA online data archives.

*Author contributions.* P.V. and A.S. conceived the experiment. N.M., R.E., P.V., A.S, H.A.K and C.T. collected the samples and ran the
experimental analyses. N.M. analyzed and interpreted the results. N.M. wrote the manuscript with inputs from all authors.

*Competing interests.* The authors declare no competing financial interests.

*Acknowledgements.* The RECAP ice coring effort was financed by the Danish Research Council through a Sapere Aude grant, the NSF through the Division of Polar Programs, the Alfred Wegener Institute, and the European Research Council under the European Community's Seventh Framework Programme (FP7/2007-2013) / ERC grant agreement 610055 through the Ice2Ice project and the Early Human Impact
project (267696). This study has also received funding from the European Research Council Executive Agency under the European Union's Horizon 2020 Research and innovation programme (Project 'ERC-2016-COG 726349 CLIMAHAL'). We thank our colleagues from the Continuous Flow Analysis group at Centre for Ice and Climate (Copenhagen, Denmark) for helping in the sample collection. We thank Trevor Popp, Steffen Bo Hansen and the RECAP drilling group, Joel Pedro, Markus Jochum and Trond Dokken for discussions. We thank Zhao Meixun and Francesco Muschitiello for sharing the Icelandic and Norwegian core data. We thank Giovanni Baccolo for his useful
insights.

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

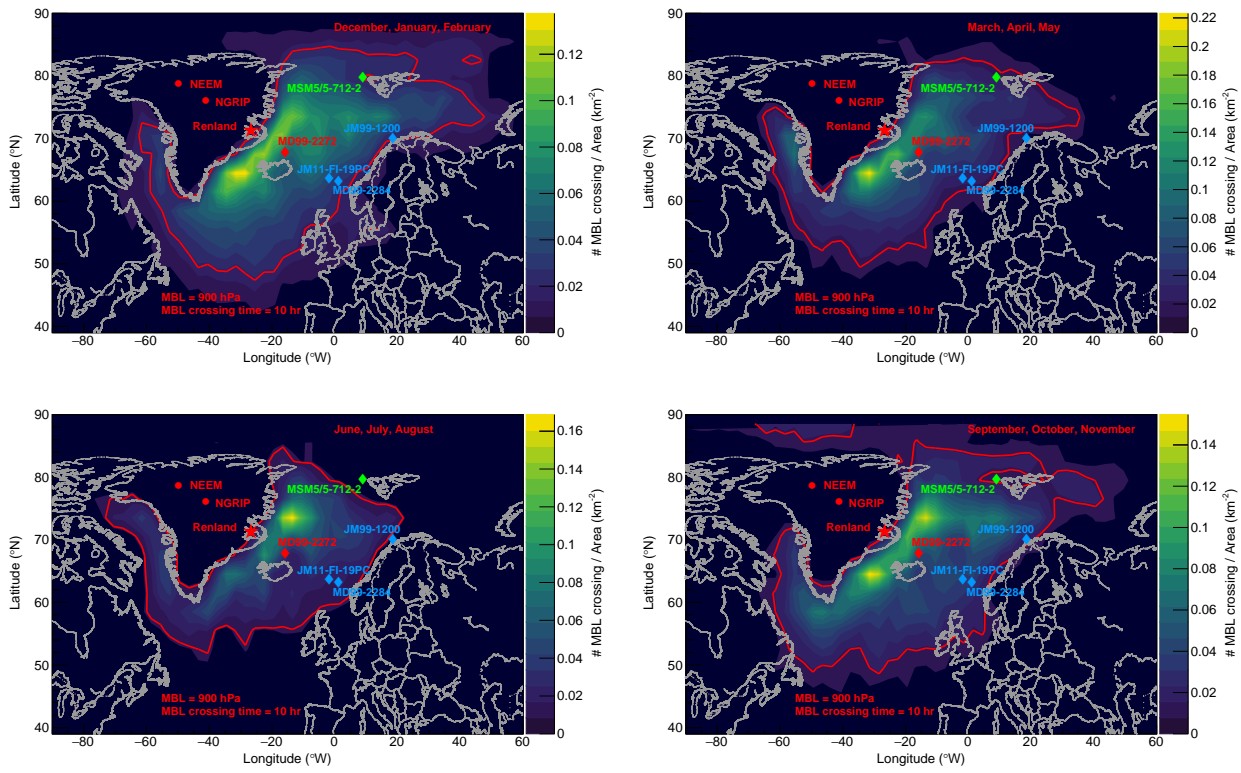

**Figure 1.** Modern (2000-2016) source region of atmospheric sodium and bromine deposited at the Renland site estimated from 3-day back trajectory calculations. On the z-axis is represented the number of Marine Boundary Layer (MBL) crossings per unit area, derived by selecting only those trajectories which crossed the 900 hPa isosurface for at least 10 hours (see Sect. 2.3). The red line contours 75% of the distribution integral and it is considered to be the bromine and sodium source region. It covers the North Atlantic Ocean from ∼50° N to 85° N in latitude. A minor contribution is expected from West Greenland waters. Note that areas outside the 75% line with counts <0.02 crossings km$^{-2}$ are not coloured. The analysis is performed on a seasonal basis. The Renland ice cap is marked with a star; the NEEM and NGRIP core sites are marked with a circle. The diamonds indicate the marine cores discussed in the text: Svalbard margin (MSM5/5-712-2, Müller and Stein 2014); North Icelandic shelf (MD99-2272, Xiao et al. 2017) and the Norwegian Sea (JM11-FI-19PC, Hoff et al. 2016; MD99-2272, Muschitiello et al. 2019; JM99-1200, Cabedo-Sanz et al. 2013).

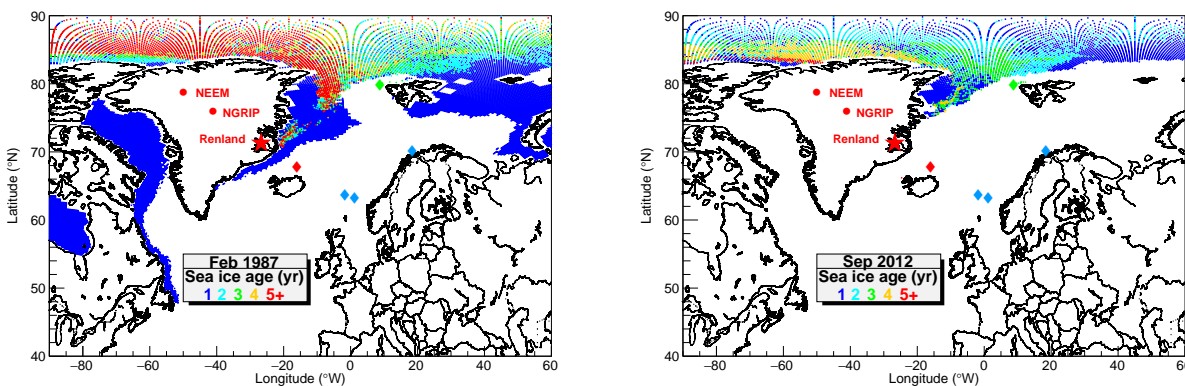

**Figure 2.** Map of modern Arctic sea ice age shown in representative extreme conditions: winter maximum (February 1987, left) and summer minimum (September 2012, right). The marine cores discussed in the text are indicated with colored diamonds. See Fig. 1 for the core names and references. The sea ice age data are from NSIDC/NASA (v.3) (Tschudi et al., 2016).

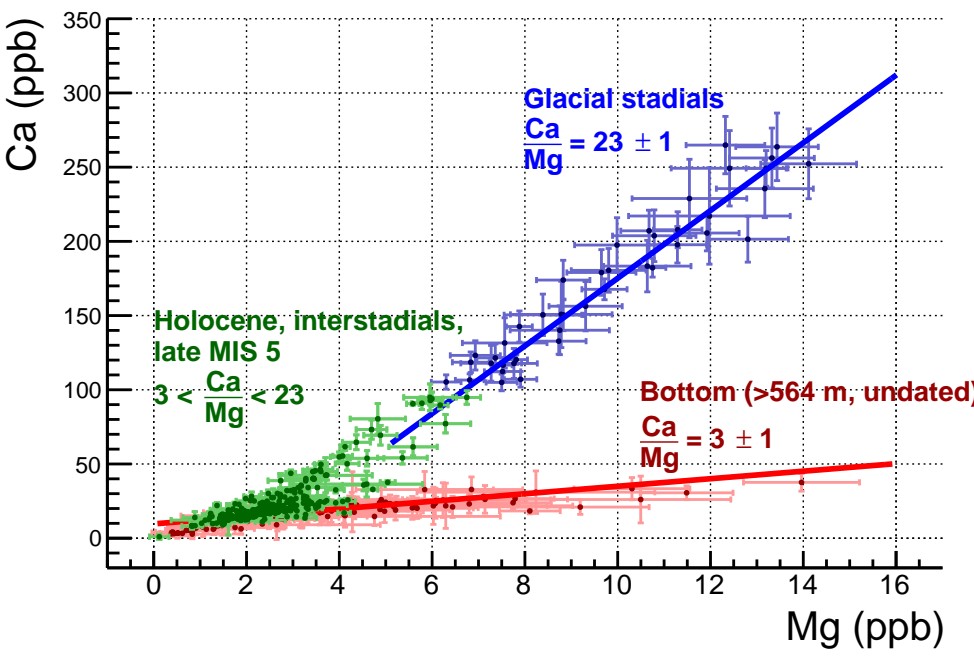

**Figure 3.** Calcium and magnesium concentrations in the RECAP core. Samples with high (>100 ppb) calcium concentrations are drawn in blue. The red samples of the bottom 22 meters (562–584 m depth) are not dated but may consist of Eemian ice. All others samples are drawn in green. The Ca/Mg ratio varies between $3\pm1$ (Asian dust) and $23\pm1$ (gypsum and/or excess of carbonate). The sea water ratio $(Ca/Mg)_m=0.32$ cannot be identified since both elements are dominated by terrestrial contributions.

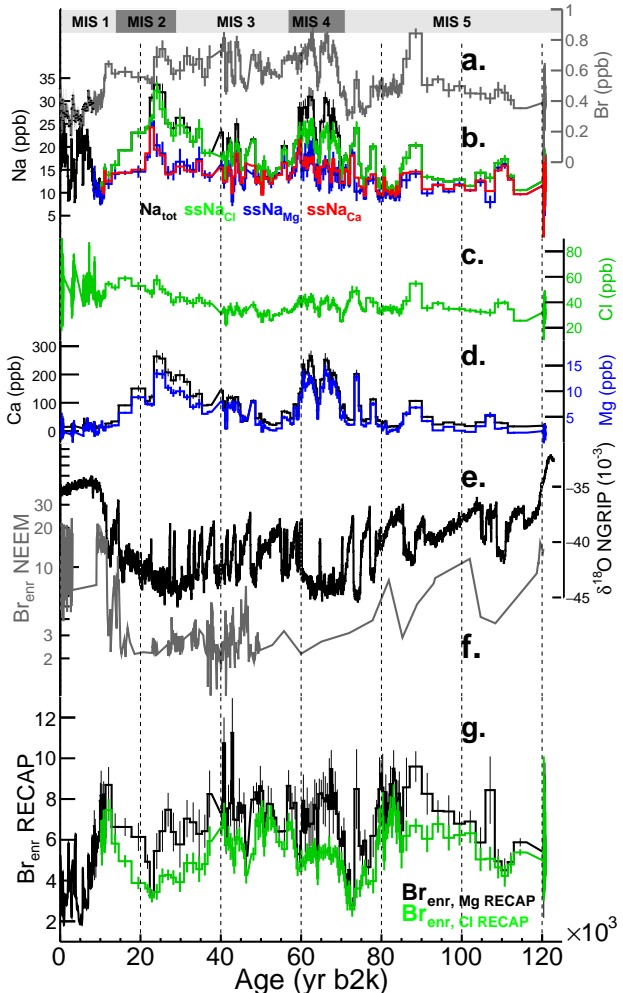

**Figure 4.** 120 kyr time series of analyte concentrations in the RECAP ice core. The last 8 kyr are 100 year averages. (a) bromine; (b) total sodium and sea-salt sodium (ssNa) series calculated using respectively chlorine (green), magnesium (blue) and calcium (red); (c) chlorine; (d) calcium (black) and magnesium (blue); (e) NGRIP $\delta^{18}$O from North Greenland Ice Core Project members (2004); (f) NEEM $Br_{enr}$ from Spolaor et al. (2016b); (g) RECAP $Br_{enr}$ series ($Br_{enr,Cl}$, green and $Br_{enr,Mg}$, black) calculated using chlorine and magnesium respectively.

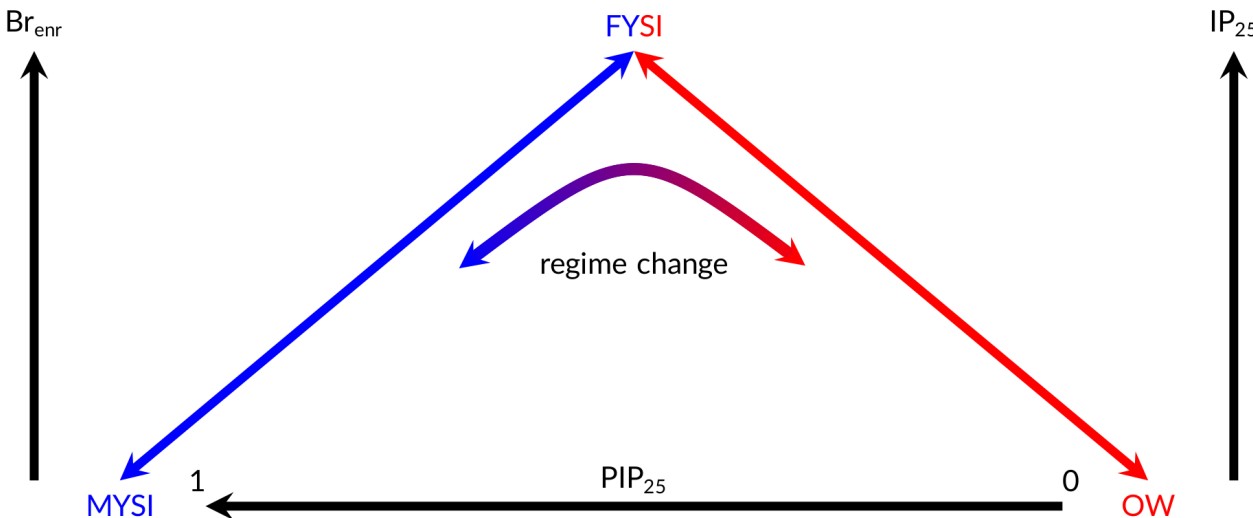

**Figure 5.** Schematic illustration of marine and (suggested) ice core-based sea ice indicators as a function of different sea ice conditions. $Br_{enr}$ (ice core) and $IP_{25}$ (sediment core) increase as a function of first-year sea ice (FYSI) area. Low values can thus indicate either open water (OW) or multi-year sea ice (MYSI) conditions, depending on the climate state. The $PIP_{25}$ index is a marine-derived semi-quantitative indicator of local sea ice conditions at the marine core location. $PIP_{25}\sim0$ ($PIP_{25}\sim1$) indicates open water (perennial sea ice, i.e. MYSI) conditions, while intermediate $PIP_{25}$ values reflect seasonal sea ice (i.e. FYSI). $Br_{enr}$ values in the RECAP core are believed to indicate a FYSI/OW regime in a 'warm' climate state and a FYSI/MYSI in a 'cold' climate state.

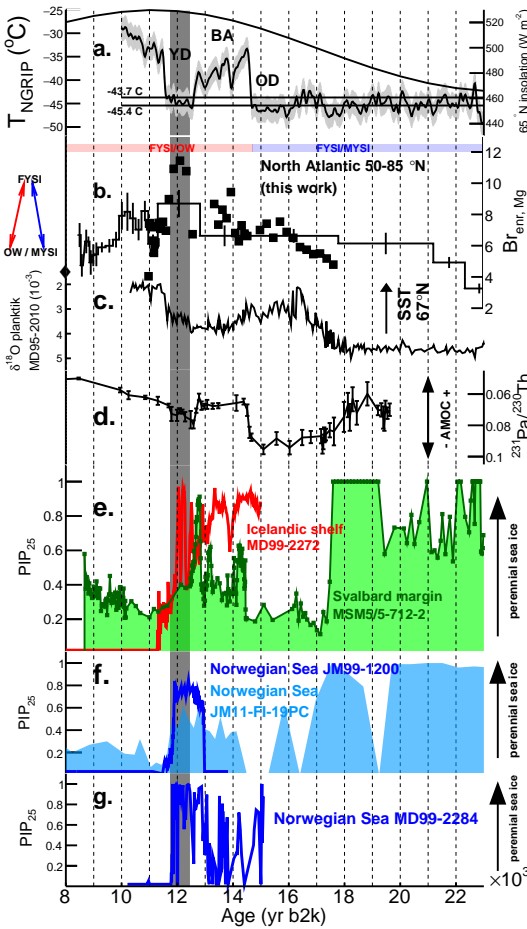

**Figure 6.** Climate records during the last deglaciation. (a) NGRIP air temperature reconstruction (Kindler et al., 2014) ($\pm 1\sigma$ = 2.5 °C). The two horizontal lines correspond to a $\pm 2\sigma$ deviation from the mean temperature (-44.6 °C) found during the 12.4-11.8 kyr period of maximum $Br_{enr,Mg}$ (grey vertical band). The $T_{NGRIP}$ record does not extend before 10 kyr but the temperature is not expected to cross the threshold anytime in the Holocene. The black line is the 21 June daily mean insolation at 65 °N (Laskar et al., 2004). (b) RECAP $Br_{enr,Mg}$ (thick black line) and repeat sampling conducted at higher resolution (squares). Since for the high-resolved measurements magnesium was not measured, for the calculation of the sea-salt sodium concentrations and the $Br_{enr,Mg}$ values (Eq. 1-7) we used the magnesium values of the 4 low-resolution samples. Over the time period covered by the repeat measurements, nssNa $\simeq$ 10-20%. The average $Br_{enr,Mg}$ value measured during the Holocene is indicated by a diamond. (c) Planktonic $\delta^{18}O$ record from sediment core MD95-2010 (66° 41.05' N; 04° 33.97' E, from Dokken and Jansen 1999). Lower $\delta^{18}O$ values indicate warmer and fresher ocean waters. (d) Record of $^{231}Pa/^{230}Th$ ratios within a sediment core retrieved in the deep western subtropical Atlantic (33° 42' N 57° 35' W). Sedimentary $^{231}Pa/^{230}Th$ are believed to reflect Atlantic meridional overturning circulation (AMOC) strength (McManus et al., 2004). (e, f, g) $PIP_{25}$ records from the Svalbard margin (green, core MSM5/5-712-2, Müller and Stein 2014), the North Icelandic shelf (red, core MD99-2272, Xiao et al. 2017), and the Norwegian Sea (core JM11-FI-19PC from Hoff et al. 2016), core JM99-1200 from Cabedo-Sanz et al. 2013 and core MD99-2284 from Muschitiello et al. 2019). The $PIP_{25}$ scale varies from perennial sea ice ($PIP_{25} \approx 1$) to open water ($PIP_{25} \approx 0$) conditions (see Fig. 5). See Fig. 1 for the $PIP_{25}$ core locations.

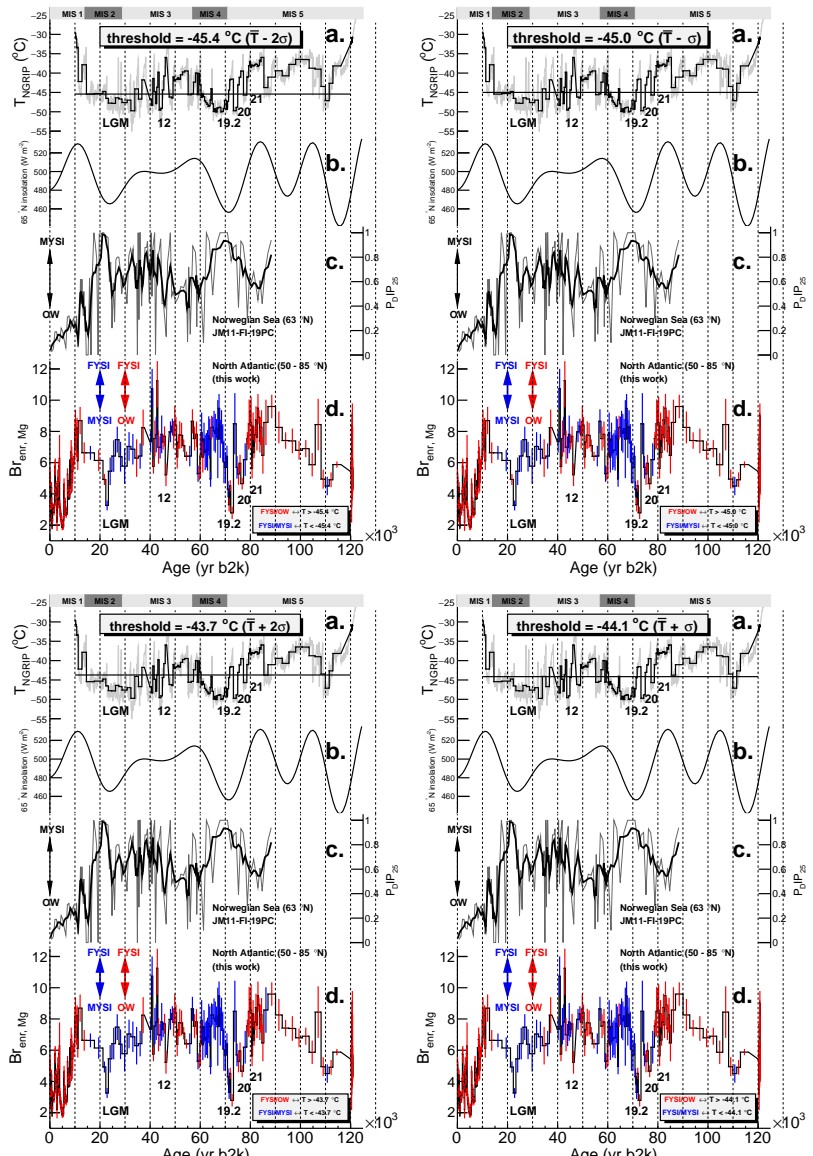

**Figure 7.** Discrimination of the two $Br_{enr,Mg}$ regimes according to the NGRIP temperature threshold (see Fig. B1 for the equivalent analysis using $Br_{enr,Cl}$). a) reconstructed NGRIP temperature (grey, Kindler et al. 2014). The temperature threshold value is the mean temperature ($\overline{T} \pm 2\sigma$: left; $T \pm 1\sigma$: right) during the 11.8-12.4 kyr period, when the change of $Br_{enr,Mg}$ regime was observed (see Fig. 6). The black time series is the temperature profile downscaled to the measured $Br_{enr}$ resolution. The numbers indicate the Greenland Interstadials (GI) discussed in the text. b) 21 June daily mean insolation at 65 °N (Laskar et al., 2004). c) 90 kyr $P_D IP_{25}$ record from the Norwegian Sea (grey, from Hoff et al. 2016) and 5-point running mean (black). d) discrimination of the $Br_{enr,Mg}$ regimes computed according to the integrated temperature value with respect to the threshold. The FYSI/OW and FYSI/MYSI regimes are indicated with red and blue error bars respectively.

**Appendix A:  Modern Na/Mg and Na/Ca ratios in dust from Northern Asian deserts**

The aim of this section is to obtain a reference value for the present day mass ratios of Na/Mg and Na/Ca in Asian dust, since the Gobi and desert regions of Mongolian and northern China (hereafter referred to as Asian dust) is believed to be the main source region of dust found in Greenlandic ice cores during the glacial period, as inferred from Sr, Nd and Pb isotope composition (Biscaye et al., 1997).

We consider n=6 published studies in which the chemical composition of dust sampled during "Desert Events" (DS) was analyzed in China. Often these studies are carried out in the framework of pollution-related research. We also include composition data from dust sampled in-situ in desert areas. For details on the single studies, we refer the reader to the references in Table A1. We note that our compilation only considers dust samples originated (or sampled directly) from deserts. Data from trajectories that traveled over other areas (e.g. marine or others) are not considered.

By considering all the above-mentioned studies, the mean Na/Mg mass ratio is 1.23 (median = 1.26). The mean Na/Ca mass ratio is 0.38 (median = 0.33). The mean Ca/Mg mass ratio is 3.66 (median = 3.55).

**Ta et al. (2003)** — Results of the X-ray fluorescence analysis of soil samples from the desert/Gobi areas in Gansu province, China

| | Desert Areas | | | | Gobi Areas | | | | |
| --- | --- | --- | --- | --- | --- | --- | --- | --- | --- |
| - | Dunhuang | Yumen | Anxi | Zhangye | Linze | Wuwei | Gulang | Yumen | Anxi |
| Na$_2$O (% ox) | 1.97 | 2.05 | 1.19 | 1.46 | 1.76 | 1.59 | 1.47 | 1.36 | 2.62 |
| MgO (% ox) | 1.96 | 2 | 3 | 1.42 | 1.38 | 1.05 | 0.91 | 1.16 | 1.3 |
| CaO (% ox) | 6.5 | 6.31 | 10.59 | 2.21 | 2.95 | 2.23 | 2.23 | 8.4 | 4.88 |
| Na (ppm) | 14615 | 15208 | 8828 | 10831 | 13057 | 11796 | 10905 | 10089 | 19437 |
| Mg (ppm) | 11821 | 12062 | 18093 | 8564 | 8323 | 6333 | 5488 | 6996 | 7840 |
| Ca (ppm) | 46456 | 45098 | 75687 | 15795 | 21084 | 15938 | 15938 | 60035 | 34877 |
| Na/Mg | 1.24 | 1.26 | 0.49 | 1.26 | 1.57 | 1.86 | 1.99 | 1.44 | 2.48 |
| Na/Ca | 0.31 | 0.34 | 0.12 | 0.69 | 0.62 | 0.74 | 0.68 | 0.17 | 0.56 |
| Ca/Mg | 3.93 | 3.74 | 4.18 | 1.84 | 2.53 | 2.52 | 2.90 | 8.58 | 4.45 |

**Wang et al. (2018)** — Results of Total Suspended Particle elemental concentrations in samples collected at Fudan University (Shanghai, China) during the 19 to 22 March 2010 dust storm estimated to originate from the Gobi Desert.

| - | 19 Night | 20 Day | 20 Night | 21 Day | 21 Night |
| --- | --- | --- | --- | --- | --- |
| Na ($\mu$g m$^{-3}$) | 1.3 | 7.9 | 19.1 | 2.8 | 3.9 |
| Mg ($\mu$g m$^{-3}$) | 1.4 | 9.5 | 14.4 | 2.2 | 2.3 |
| Ca ($\mu$g m$^{-3}$) | 8.5 | 37.3 | 56.9 | 9.4 | 8.6 |
| Na/Mg | 0.93 | 0.83 | 1.33 | 1.27 | 1.70 |
| Na/Ca | 0.15 | 0.21 | 0.34 | 0.30 | 0.45 |
| Ca/Mg | 6.07 | 3.93 | 3.95 | 4.27 | 3.74 |

**Sun et al. (2005)** — Elemental concentrations in aerosols sampled during two springtime 2002 Desert Storms that hit Beijing, both of which originated from North Asia deserts.

| - | Desert Storm 1 | Desert Storm 2 |
| --- | --- | --- |
| Na ($\mu$g m$^{-3}$) | 37 | 39 |
| Mg ($\mu$g m$^{-3}$) | 34 | 29 |
| Ca ($\mu$g m$^{-3}$) | 178 | 58 |
| Na/Mg | 1.09 | 1.34 |
| Na/Ca | 0.21 | 0.67 |
| Ca/Mg | 5.24 | 2.00 |

**Fan et al. (2013)** — Elemental concentrations measured in Guangzhou (China) during the 23-25 April, 2009 dust storm originated in North China.

| - | Desert Storm |
| --- | --- |
| Na ($\mu$g m$^{-3}$) | 2.98 |
| Mg ($\mu$g m$^{-3}$) | 2.83 |
| Ca ($\mu$g m$^{-3}$) | 5.78 |
| Na/Mg | 1.05 |
| Na/Ca | 0.52 |
| Ca/Mg | 2.04 |

**Zhang et al. (2002)** — Composition of air-volume-based concentration of aerosols sampled in Minquin (northwest China desert region) in March–May 1995 and April–June 1996.

| - | Minquin 1995 | | Minquin 1996 | |
| --- | --- | --- | --- | --- |
| - | Geometric mean | Average | Geometric mean | Average |
| Na ($\mu$g m$^{-3}$) | 1.37 | 3.22 | 1.24 | 1.8 |
| Mg ($\mu$g m$^{-3}$) | 2.14 | 4.4 | 2.07 | 2.88 |
| Ca ($\mu$g m$^{-3}$) | 5.7 | 11.4 | 6.21 | 8.41 |
| Na/Mg | 0.64 | 0.73 | 0.60 | 0.63 |
| Na/Ca | 0.24 | 0.28 | 0.20 | 0.21 |
| Ca/Mg | 2.66 | 2.59 | 3.00 | 2.92 |

**Wang et al. (2009)** — Arithmetic mean elemental concentrations measured in n=317 dust fallout samples collected from 33 sites located in the Gobi Desert, Sandy Desert, Loess, and steppes in North China from April 2001 to March 2002.

| | |
| --- | --- |
| Na (%) | 1.47 |
| Mg (%) | 1.01 |
| Ca (%) | 3.40 |
| Na/Mg | 1.46 |
| Na/Ca | 0.43 |
| Ca/Mg | 3.37 |

Table A1: compilation Na/Mg, Na/Ca and Ca/Mg mass ratios in dust samples originated from North Asia deserts.

**Appendix B:  Discrimination of the two $Br_{enr,Cl}$ regimes according to the NGRIP temperature threshold**

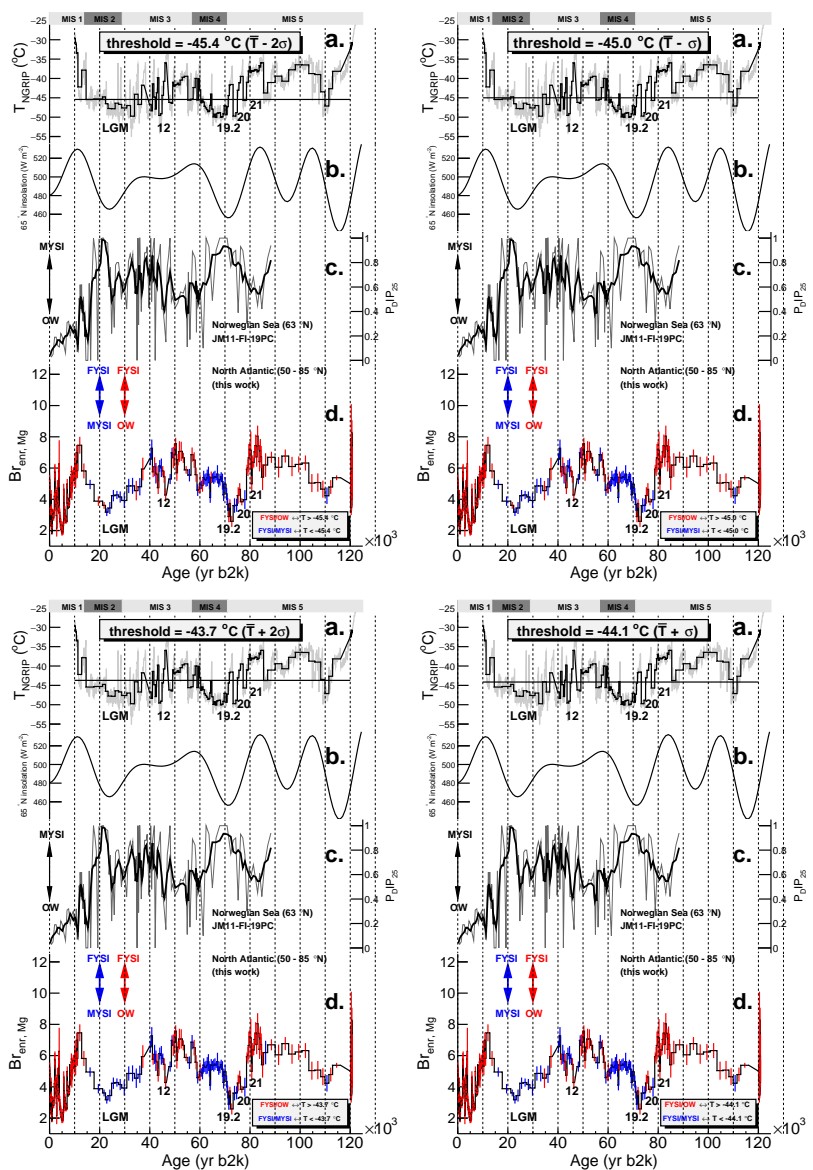

**Figure B1.** Discrimination of the two $Br_{enr,Cl}$ regimes according to the NGRIP temperature threshold. This figure is equivalent to Fig. 7 with $Br_{enr,Mg}$ replaced by $Br_{enr,Cl}$.