# Peer review of "120,000 year record of sea ice in the North Atlantic?"

_Climate of the Past, 2018_

## Referee Comment (RC1) · Anonymous Referee #1 · 10 Aug 2018

This paper presents a new 120,000 year record of Br measured in the Renland ice core. It then interprets this in terms of sea ice extent. The analytical record is an interesting and potentially important one, and the paper should eventually become publishable in Climate of the Past. It is written in a clear style. However there are a number of issues that need to be addressed before it can be published. These fall principally into three areas, the meaning of Br_enr, dating of the ice, and the transformation the authors adopt.

1. The idea that some measure of Br enrichment may be associated with mainly first year sea ice (FYSI) is plausible and the authors have, in previous papers, made some kind of argument for it. However it is far from established and the paper is far too definite about that. It needs to spell out the caveats, and use the word "suggest", "might" and "may" a lot more. I would also argue to add a question mark to the title.

The idea that Br_enr is a FYSI proxy relies on a number of assumptions.

(a) The Br activation process leads to production of activated Br and to a depletion of Br in the salty surface on which it happens. As a result there can be both enhancements and depletions in what is deposited as snowfall, depending on the relative importance of depleted sea salt aerosol and gas-phase Br compounds that eventually get transported as (presumably) HBr. The implicit assumption used here is that the gas phase Br is transported to (in this case) Renland much more efficiently than the depleted sea salt aerosol. I agree with this, which is consistent with eg Simpson et al 2004. However the assumption needs to be stated, especially as thi group has been inconsistent on that question, with Spolaor et al (2013), the first paper on this proxy, asserting exactly the opposite.

(b) There is also an assumption that HBr is produced only by the photochemical Br activation and not by a reaction between salt and acid analogous to that described on page 3, line 10 for Br. Again, the weak Cl fractionation suggests this may be true, but it needs stating.

(c) The paper assumes that Br activation takes place only on FYSI and not at all on MYSI. This is highly unlikely to be true, and the production ratio between these two forms of sea ice will be crucial to the interpretation. This is also the case for sea salt aerosol itself as shown in a modelling study by Rhodes et al (2018), where the use of sea salt as a sea ice proxy is crucially dependent on the extent to which FY and MYSI are involved in aerosol production. I don't expect a full modelling study here, but again it needs to be acknowledged that production on MYSI would affect the interpretation.

(d) Finally the authors essentially assume that their untransformed proxy equates to FYSI, taking no account of transport distance. It's actually quite plausible that their proxy decreases in very cold periods not because there is less FYSI but because it's much further from Renland. This may still argue for a similar interpretation to the data, but is a more subtle mechanistic point that will eventually need to be tested in models.

I therefore ask the authors to state their assumptions and the caveats more fully, and I will suggest some places in the text where the interpretation should be more tentative.

2. This seems to be the first paper in which RECAP data have been presented covering the glacial, and is therefore the one in which the age model is effectively first presented. Dating Renland is far from simple, and it is completely unacceptable to rely on a reference to an in preparation paper (Simonsen) which the reader has no sight of. If this is the first paper

published then this one needs to demonstrate that the dating is plausible – in the appendix or SI I would expect at minimum an age-depth profile, a table of tie points, and an estimate of age uncertainty. Without that, the reader has no idea how robust the comparisons to the NGRIP isotope record are (ideally we would see the Renland water isotope record in this paper but I appreciate that must be the subject of another paper and that this paper is out of sequence). In the absence of a substantial dating appendix/SI, this paper should not be published until a full description of the dating has been published in another paper.

3. While it is a clever idea, I am not convinced by the value of the transformation. The paper claims that this "linearizes" the record, but how does it do that? The number that is achieved has not in any sense been shown to be linear with any climate variable, be it FYSI extent, total sea ice extent or anything else. What has been done is to give a spurious air of quantitativeness to a proxy that remains at present qualitative. In Fig 5 the panel where the data are colour coded red or blue is good, and I understand the temptation to try to turn it into a curve that can be compared with the IP25 record in Figure 6 but I feel it should be resisted. If the authors cannot resist producing something I suggest a different alternative: (a) Have the y-axis go up to 7.4 (or whatever the change point is considered to be) and then decreases back to zero again, and plot the actual Br_enr value either above or below the line, maintaining its colour coding . (Mathematically this is achieved by making eq5 simply Br_enr=-Br_enr, plotting it with axis reversed and using the absolute value of Br_enr as the axis label). To indicate the uncertainty about the T threshold however, you should plot both the actual and the transformed Br_enr values for data within 2 sigma of the chosen temperature threshold so that it is obvious that there are two possible states at some depths. This procedure will produce a curve similar to the one in Figure 6 but without producing spurious new numbers.

More detailed comments

Page 1, abstract, needs to be more tentative: line 3 I suggest "and tentatively reconstruct"; line 6 "what we interpret as the transition from MYSO to FYSI started at 17.6 kyr"; line 8 "our proxy interpreted as FYSI reached its maximum"; line 10 "sea ice extent was probably greatest".

Page 2, line 1 "reservoirs"

Page 3, equations 2-3 and surrounding text. I appreciate that correcting for crustal Na in Greenland is not straightforward, but you need to explain what your rationale is here better, and when you do, it doesn't completely make sense. You are implicitly assuming that when the salt/acid reaction has created excess Cl, there is no terrestrial Na (eq 3); and that if there is a Cl/Na ratio less than 1.8 that is entirely because of terrestrial Na, and not at all because of the salt/acid reaction leading to removal of Cl from aerosol. I am not sure why either of those assumptions should be correct. Normally one would use a crustal element such as Al or perhaps Ca to put limits on the terrestrial correction. In the absence of that you need to revisit your method and at minimum state its limitations and suggest how much it may affect your results.

Page 3, line 15-20. Thank you for using the word "suggesting" here. However I am not sure what point you are making in line 19-20. Why would we expect Holocene-like ice extent at 120 kyr when we are already well into the glacial inception?

Page 3, line 26-28. The modelling study of Rhodes et al (Rhodes, R. H. et al., GRL, 45(11), 5572-5580, doi:10.1029/2018gl077403) doesn't really support your interpretation, as in present conditions, it says that both sites are overwhelmingly seeing OW conditions (Rhodes was looking at the influence of sea salt aerosol but I imagine modelling Br would produce a similar result). Comment? The difference between Holocene values for the two sites could also be partly related to the fact that the Br_enr measure is not really ideal because it does not measure the amount of Br reaching the site but rather the ratio of Br/sea salt aerosol. NGRIP of course receives much less sea salt than Renland so just a small amount of gas phase Br can induce a Br_enr >>1. This may not fully explain the difference between the sites but it is a factor. It's too late now because this measure is embedded in the literature, but the use of Br_excess ([Br-Br_seawater] instead of [Br/Br_seawater]) would have avoided this problem.

Page 4, line 8 "a number of"

Page 6, last line "Our reconstruction suggests.."

Figure 2. I appreciate that Kindler only goes to 10 kyr but it is unhelpful that we don't see a climate record from 10-0 kyr. Can you infill with an 18O record to at least indicate to the reader that Holocene temperatures remain warm.

Fig 3 caption, Fig 6 caption: Tzedakis et al is not an appropriate reference for orbital parameters. Please cite original papers by either Berger or Laskar.

Appendix B. I am wondering why you call this "sea salt aerosol source area" when what you are interested in is the source of the gas phase Br. The back trajectories are for air masses so certainly not specific to aerosol.

Fig B1 caption line 2 "constraint" (but can you explain what you mean by this constraint).

Fig B1 caption. Sorry for my ignorance but I never heard of a nabla before, can't you call it an inverted triangle?

---

## Referee Comment (RC2) · Anonymous Referee #2 · 10 Aug 2018

The paper by Maffezzoli presents the first data on sea ice coverage of the North Atlantic and the Greenland, Iceland, Norwegian (GIN) Seas based on an ice core recently drilled on a coastal ice cap from East Greenland (Renland ice core). Following the previous approach by Spolaor et al. (2016) they use the Br enrichment above sea salt concentrations (Br_enr) linked to halogen explosions occurring on seasonal sea ice surfaces. In principle, the data and argumentation of the paper are convincing although at some points not explained in detail enough. The paper represents an important contribution to the field that will be of interest for ice core specialists, marine geologists and modelers alike.

In its current state, however, the paper still suffers from some language issues (see

annotated pdf file attached), some structural deficits and some issues with the argumentations that I outline in my general comments below. I am convinced that these changes can be accommodated and recommend to accept the paper after major revisions.

General comments

- Page 2 Line 17-23, "We compare ... Belt and Muller, 2013)": This text should only come at the end of section 2

- Section 2, 2nd paragraph: In this paragraph it is stated that the origin of Renland sea salt is the North Atlantic and the text refers to Appendix B. This is a crucial piece of information and should not be hidden in an Appendix but included in the main text. Moreover some more information on this statement would be helpful for the reader:

a) sea salt aerosol has a pronounced seasonal cycle with a broad maximum in the winter half-year. This also holds for eastern coastal sites (Oyabu et al., Polar Science, 2016). Thus, the Renland sea salt record is mainly representative for the winter half-year. The back trajectory analyses should be done both for the winter and summer half-year separately and shown in two panels. Note also that due to the limited lifetime of sea salt aerosol, short trajectories are likely to bring more sea-salt aerosol to Renland compared to longer trajectories.

b) transport pathway (trajectories) is one side of the coin, the sea ice source is the other. An additional figure/panel showing the multiyear and first year sea ice distribution in spring (at its maximum) would be helpful to support the claim that the source regions identified in the trajectory study are covered by FYSI or OW. The National Snow and Ice Data Center provides this information (https://nsidc.org/data/nsidc-0611).

- A method section is missing. Again, this is hidden in an Appendix A but should be part of the main text.

- Page 3 line 15-20, "The Br_enr ... Holocene value.": This text should be the beginning

of section 3

- Page 3 line 22-29. Here the NEEM Br_enr record is mentioned and compared to Renland. In order to allow the reader to make this comparison, the NEEM data should be shown in one of the figures!

- Page 4 line 1: Here the expression "tipping point" is used. There is no clear definition of what a tipping point is, but in climate science it is usually used for a rapid regime shift (see IPCC). The change from MYSI to FYSI to OW, however, is likely a gradual process. Accordingly, I would suggest to avoid the expression tipping point in the manuscript. Along this line, I think section 3.2 (linearization of the Br_enr record) and its application on the 120 kyr record in section 3.3 does not provide added value and in fact is misleading due to the gradual nature of the MYSI/FYSI/OW transition. Instead of trying to force this mathematically to a monotonous function, I would recommend to just use color bars in the figures underlying the records to indicate where MYSI, FYSI or OW dominate the Br_enr record. Moreover, mean Greenland temperature may not be the only parameter determining the amount of FYSI present (see next comment).

- The high resolution data presented in Figure 3 clearly show that the YD is the time period of largest Br_enr (clear maximum) during termination I, thus the largest FYSI presence. In particular Br_enr is clearly higher in the YD than in the OD period. Note that there is no similar Br_enr maximum during the OD/BA transition as seen in the YD at the point when temperature during the OD/BA transition is crossing the same temperature as found in the YD. Either this point is just missed in the record (unlikely), or NGRIP temperature alone is not able to fully explain the observations. Here additional information could be used to elaborate on this issue. First of all, FYSI is strongly dependent on the seasonal temperature variation, this should be mentioned somewhere. Models suggest (Buizert et al., Science, 2014) that temperature seasonality during the YD and OD was much higher than during the BA. This could explain why the YD has higher FYSI than the BA. The difference between YD and OD sea ice conditions (Br_enr levels) may potentially be explained by the overall much lower temperatures

encountered during the OD than in the YD (Buizert et al., 2014), which may push the OD sea ice regime toward more MYSI. This difference may be linked to the generally higher AMOC strength in the YD compared to the OD (McManus et al., Nature 2004).

- Page 4 line 29-33: Here the paper by Rasmussen is referred to, but I am not sure - based on the text provided - whether it is referred to correctly and whether the statement made in the manuscript is correct. Rasmussen et al. (2016) claim that in the North Atlantic south of Iceland SST warming already starts during stadial conditions, while in the GIN seas the warming starts only with the Greenland DO onset, i.e., when sea ice rapidly declines. Rasmussen et al., do not explicitly discuss the YD/BA/OD transitions and in fact their record does not show a clear early warming during the reduced AMOC conditions of the YD and OD. Accordingly, to make this statement would require high resolution Br_enr data for selected DO events from the Renland record, which are not available yet. I would suggest to remove this statement and also the reference to mean ocean temperatures, Antarctica and CO2.

- Page 6, line 26. Here you refer to the GI numbers. These should be included in the figures

- Include Fig. 1 in Fig. 2

- Add the NEEM record in Fig. 2 or provide a separate figure for the NEEM/Renland comparison. Add color bars for sea ice conditions underlying figure 2

- add color bars for sea ice conditions underlying figure 3

- remove figure 5

- remove the transformed BR_enr in figure 6, add color bars

- move Appendix A to a Method section

- move Appendix B to section 2 and add more information as outlined above

Specific Comments

- see annotated pdf

Please also note the supplement to this comment:
https://www.clim-past-discuss.net/cp-2018-80/cp-2018-80-RC2-supplement.pdf

**Supplement:**

**120,000 year record of sea ice in the North Atlantic**

Niccolò Maffezzoli1, Paul Vallelonga1, Ross Edwards2,3, Alfonso Saiz-Lopez4, Clara Turetta5,6, Helle Astrid Kjær1, Carlo Barbante5,6, Bo Vinther1, and Andrea Spolaor5,6

[revised manuscript text omitted]

---

## Referee Comment (RC3) · Anonymous Referee #3 · 22 Aug 2018

As the first two reviewers already raised the main concerns of the manuscript for which I totally agree, I will only focus on a general comment.

Throughout of the manuscript the authors take granted that the enrichment of Br above sea water has only one explanation, namely the first-year sea ice extension. The discussion that follows never question this hypothesis and only interprets enriched Br as the first-year sea ice extension. Such concordance is so strong in the paper that enriched Br and first-year sea ice extension are used indifferently, like synonyms.

The use in ice core/snow of Br-excess as a proxy of sea ice extent was initiated in 2013 by the present author's team. Since then, this team has published around 10 papers with most of them using the same adequacy Br-excess = FYSI to reconstruct sea ice in Russian & Canada Arctic, East Antarctica, in East Greenland.

[Figure]

However, quoting their own conclusion in their 2013 original paper: "Although further investigation is required to characterize potential depositional and post-depositional processes, these preliminary findings suggest that I and Br can be linked to variability in the spring maximum sea ice extension and seasonal sea ice surface area (Spolaor et al., The Cryosphere, 2013)". Requirement and suggesting are the two most important terms in their conclusions. With time "suggest" has become a certitude and "require" has turned into still awaiting. I could not find in ISI database any papers treating this hypothesis at the process level as originally suggested the authors.

In J Abbath's Nat Geos comment of Pratt et al. 2013 paper (which results are at the heart of their hypothesis), I quote "Whether the rising fraction of young sea ice will enhance snowpack bromine production and release, and concomitant changes in atmospheric chemistry, remains to be seen (J. Abbath, Nat Geo, 2013)". Interestingly, Pratt's paper argues that acidity of snow is a prerequisite for Br activation as well as internal snow pack air chemistry, e.g OH snow pack production, nitrate/nitrite concentration etc. As the relative scavenging efficiency between ssNa and gaseous Br seems to be the main driver of the Br-excess, change in precipitation regime can potentially greatly influence the Br/Na ratio. A transect study from coast the inland along a strong accumulation gradient will be very informative in this view. The speciation of Bry species, air snow transfer are also important parameters to consider. In summary, the current literature gives little concrete elements to definitely link Br-excess and FYSI extension. Increase production efficiency at constant extension is for instance never considering, neither change in scavenging precipitation, etc. A correct scientific approach should be:

1- Establish the hypothesis 2- Test it against observations, determine the sensitivity to parameters 3- And use it within its limits

It seems to me that the step 2 is currently missing for Br-excess. There is a long list of acclaimed proxies in the ice core community that after a throughout investigation turned out to have been interpreted too simply (just few of them MSA, MSA/sulfate

ratio, water isotopes, levoglucosan, nitrate concentration).

In conclusion, the authors should clearly state that their hypothesis is not yet fully demonstrated and their conclusions are only hypothetical. Currently, other interpretations are possibles.

Pratt, K. A., Custard, K. D., Shepson, P. B., Douglas, T. A., Pohler, D., General, S., Zielcke, J., Simpson, W. R., Platt, U., Tanner, D. J., Gregory Huey, L., Carlsen, M., and Stirm, B. H.: Photochemical production of molecular bromine in Arctic surface snowpacks, Nature Geosci, 6, 351-356, 10.1038/ngeo1779, 2013.

---

## Author Comment (AC3) · 30 Jul 2019

REVIEW – REFEREE#3

As the first two reviewers already raised the main concerns of the manuscript for which I totally agree, I will only focus on a general comment. Throughout of the manuscript the authors take granted that the enrichment of Br above sea water has only one explanation, namely the first-year sea ice extension. The discussion that follows never question this hypothesis and only interprets enriched Br as the first-year sea ice extension. Such concordance is so strong in the paper that enriched Br and first-year sea ice extension are used indifferently, like synonyms.

The use in ice core/snow of Br-excess as a proxy of sea ice extent was initiated in 2013 by the present author's team. Since then, this team has published around 10 papers with most of them using the same adequacy Br-excess = FYSI to reconstruct sea ice in Russian & Canada Arctic, East Antarctica, in East Greenland.

However, quoting their own conclusion in their 2013 original paper: "Although further investigation is required to characterize potential depositional and post-depositional processes, these preliminary findings suggest that I and Br can be linked to variability in the spring maximum sea ice extension and seasonal sea ice surface area (Spolaor et al., The Cryosphere, 2013)". Requirement and suggesting are the two most important terms in their conclusions. With time "suggest" has become a certitude and "require" has turned into still awaiting. I could not find in ISI database any papers treating this hypothesis at the process level as originally suggested the authors.

In J Abbath's Nat Geos comment of Pratt et al. 2013 paper (which results are at the heart of their hypothesis), I quote "Whether the rising fraction of young sea ice will enhance snowpack bromine production and release, and concomitant changes in atmospheric chemistry, remains to be seen (J. Abbath, Nat Geo, 2013)". Interestingly, Pratt's paper argues that acidity of snow is a prerequisite for Br activation as well as internal snow pack air chemistry, e.g OH snow pack production, nitrate/nitrite concentration etc. As the relative scavenging efficiency between ssNa and gaseous Br seems to be the main driver of the Br-excess, change in precipitation regime can potentially greatly influence the Br/Na ratio. A transect study from coast the inland along a strong accumulation gradient will be very informative in this view. The speciation of Bry species, air snow transfer are also important parameters to consider. In summary, the current literature gives little concrete elements to definitely link Br-excess and FYSI extension. Increase production efficiency at constant extension is for instance never considering, neither change in scavenging precipitation, etc. A correct scientific approach should be: 1-Establish the hypothesis 2- Test it against observations, determine the sensitivity to parameters 3- And use it within its limits It seems to me that the step 2 is currently missing for Br-excess. There is a long list of acclaimed proxies in the ice core community that after a throughout investigation turned out to have been interpreted too simply (just few of them MSA, MSA/sulfate ratio, water isotopes, levoglucosan, nitrate concentration).

In conclusion, the authors should clearly state that their hypothesis is not yet fully demonstrated and their conclusions are only hypothetical. Currently, other interpretations are possibles.

We thank Reviewer#3 for her/his comment. We agree on the general point that the uncertainties related to the Brenr have to be clearly stated. The manuscript introduction now focuses on Brenr as a potential indicator of past sea ice conditions (*1. Introduction: Brenr as a potential indicator for past sea ice conditions*). The uncertainties and the key aspects that require future studies and investigations are

discussed. This paragraph includes both aspects that Reviewer#3 and Reviewer#1 pointed out, as well as other few aspects. This section is structured into three general aspects related to Brenr: activation, transport and deposition/postdeposition. The reader can find this section attached in the answer file to Reviewer#1 (pp2,3).

---

## Author Response (AR1)

**List of relevant changes**

- An introduction on Brenr as a potential proxy for past FYSI conditions has been added (Sect. 1).
- The whole experimental part together with the atmospheric modeling part are now presented in the Methods section (Sect. 2).
- Following the comment of Reviewer#1, the section on the Brenr calculation (Sect. 3) has been expanded and an additional analysis is performed. It now consists on 3 methods for the calculation of ssNa, based on chlorine, magnesium and calcium. Accordingly, we now include all the new measurements in the experimental description and plot all time series. On this topic, a section is added in the Appendix.
- The deglaciation section (Sect. 4.1 and Fig. 6) has been further analyzed; 2 additional marine records have been included.
- The transformation has now been removed. The general interpretation of the 120 kyr Brenr record is the same as in the previous version.

One major comment was addressed to the ice core age model, which is described in another paper. This paper has now been accepted for publication but at the moment of writing not yet published.

**REVIEW – REFEREE#1**

This paper presents a new 120,000 year record of Br measured in the Renland ice core. It then interprets this in terms of sea ice extent. The analytical record is an interesting and potentially important one, and the paper should eventually become publishable in Climate of the Past. It is written in a clear style. However there are a number of issues that need to be addressed before it can be published. These fall principally into three areas, the meaning of Br\_enr, dating of the ice, and the transformation the authors adopt.

We thank the Reviewer for the time she/he took to review the manuscript. We agree with her/his comments and incorporated the suggested modifications, except for the ice core chronology, which is still not published. In particular, the introduction (Sect. 1.) is centered on the meaning/limitations of Brenr. The transformation has been removed. An extended section (Sect. 3.) has been added to deal with the calculation of ssNa (sea-salt sodium) concentrations, needed to calculate Brenr. Now three methods are used: chlorine, magnesium and calcium. The analytical records of this paper now include bromine, sodium, calcium, magnesium and chlorine. All time series are presented in the figures in the main text and the measurements are explained in the Methods section (Sect. 2.2). Two marine PIP records from the Norwegian Sea have been added.

1. The idea that some measure of Br enrichment may be associated with mainly first year sea ice (FYSI) is plausible and the authors have, in previous papers, made some kind of argument for it. However it is far from established and the paper is far too definite about that. It needs to spell out the caveats, and use the word "suggest", "might" and "may" a lot more. I would also argue to add a question mark to the title.

The idea that Br\_enr is a FYSI proxy relies on a number of assumptions.

(a) The Br activation process leads to production of activated Br and to a depletion of Br in the salty surface on which it happens. As a result there can be both enhancements and depletions in what is deposited as snowfall, depending on the relative importance of depleted sea salt aerosol and gas-phase Br compounds that eventually get transported as (presumably) HBr. The implicit assumption used here is that the gas phase Br is transported to (in this case) Renland much more efficiently than the depleted sea salt aerosol. I agree with this, which is consistent with eg Simpson et al 2004. However the assumption needs to be stated, especially as this group has been inconsistent on that question, with Spolaor et al (2013), the first paper on this proxy, asserting exactly the opposite.

(b) There is also an assumption that HBr is produced only by the photochemical Br activation and not by a reaction between salt and acid analogous to that described on page 3, line 10 for Br. Again, the weak Cl fractionation suggests this may be true, but it needs stating.

(c) The paper assumes that Br activation takes place only on FYSI and not at all on MYSI. This is highly unlikely to be true, and the production ratio between these two forms of sea ice will be crucial to the interpretation. This is also the case for sea salt aerosol itself as shown in a modelling study by Rhodes et al (2018), where the use of sea salt as a sea ice proxy is crucially dependent on the extent to which FY and MYSI are involved in aerosol production. I don't expect a full modelling study here, but again it needs to be acknowledged that production on MYSI would affect the interpretation.

(d) Finally the authors essentially assume that their untransformed proxy equates to FYSI, taking no account of transport distance. It's actually quite plausible that their proxy decreases in very cold periods not because there is less FYSI but because it's much further from Renland. This may still argue for a

similar interpretation to the data, but is a more subtle mechanistic point that will eventually need to be tested in models.

I therefore ask the authors to state their assumptions and the caveats more fully, and I will suggest some places in the text where the interpretation should be more tentative. The introduction (Sect. 1.) now consists of a description of Brenr as a potential sea ice proxy and the limitations and uncertainties that still remain. We thus proceed with the interpretation of the Brenr record as a record of FYSI, but it is now clearly stated that a number of uncertainties still exist. A question mark to the title has been added. Please find the added text in italics.

[revised manuscript text omitted]

2. This seems to be the first paper in which RECAP data have been presented covering the glacial, and is therefore the one in which the age model is effectively first presented. Dating Renland is far from simple, and it is completely unacceptable to rely on a reference to an in preparation paper (Simonsen) which the reader has no sight of. If this is the first paper published then this one needs to demonstrate that the dating is plausible – in the appendix or SI I would expect at minimum an age-depth profile, a table of tie points, and an estimate of age uncertainty. Without that, the reader has no idea how robust the comparisons to the NGRIP isotope record are (ideally we would see the Renland water isotope record in this paper but I appreciate that must be the subject of another paper and that this paper is out of sequence). In the absence of a substantial dating appendix/SI, this paper should not be published until a full description of the dating has been published in another paper.

For the ice core chronology we refer to the Supplementary of Simonsen et al. "East Greenland ice core dust record reveals timing of post-Eemian advance of Greenland ice sheet" (Nat. Comm. In press, 2019)

3. While it is a clever idea, I am not convinced by the value of the transformation. The paper claims that this "linearizes" the record, but how does it do that? The number that is achieved has not in any sense been shown to be linear with any climate variable, be it FYSI extent, total sea ice extent or anything else. What has been done is to give a spurious air of quantitativeness to a proxy that remains at present qualitative. In Fig 5 the panel where the data are colour coded red or blue is good, and I understand the temptation to try to turn it into a curve that can be compared with the IP25 record in Figure 6 but I feel it should be resisted. If the authors cannot resist producing something I suggest a different alternative:

(a) Have the y-axis go up to 7.4 (or whatever the change point is considered to be) and then decreases back to zero again, and plot the actual Br\_enr value either above or below the line, maintaining its colour coding . (Mathematically this is achieved by making eq5 simply Br\_enr=-Br\_enr, plotting it with axis reversed and using the absolute value of Br\_enr as the axis label). To indicate the uncertainty about the T threshold however, you should plot both the actual and the transformed Br\_enr values for data within 2 sigma of the chosen temperature threshold so that it is obvious that there are two possible states at some depths. This procedure will produce a curve similar to the one in Figure 6 but without producing spurious new numbers.

The transformation has been removed, and we kept only the records with the two regimes color coded in red and blue.

**More detailed comments**

Page 1, abstract, needs to be more tentative: line 3 I suggest "and tentatively reconstruct"; line 6 "what we interpret as the transition from MYSO to FYSI started at 17.6 kyr"; line 8 "our proxy interpreted as FYSI reached its maximum"; line 10 "sea ice extent was probably greatest". We have now written the abstract clearly stating the uncertainties still exist in the use of Brenr.

Abstract. Although it has been demonstrated that the speed and magnitude of recent Arctic sea ice decline is unprecedented for the past 1,450 years, few records are available to provide a paleoclimate context for Arctic sea ice extent. Bromine enrichment in ice cores has been suggested to indicate the extent of newly formed sea ice areas. Despite the similarities among sea ice indicators and ice core bromine enrichment records, uncertainties still exist regarding the quantitative linkages between bromine reactive chemistry and the first year sea ice surface. Here we present a 120,000 year record of bromine enrichment from the RECAP ice core, coastal East Greenland, and interpret it as a record of first-year sea ice. We compare it to existing sea ice records from marine cores and tentatively reconstruct past sea ice conditions in the North Atlantic as far north as the Fram Strait (50-85 °N). We find that during the last deglaciation, the transition from multi-year sea ice to first-year sea ice started at  $\sim$ 17.5 kyr, synchronous with sea ice reductions observed in the eastern Nordic seas and with the increase of North Atlantic ocean temperature. First-year sea ice reached its maximum at 12.4-11.8 kyr, after which open-water conditions started to dominate, as supported by sea ice records from the eastern Nordic seas and the North Icelandic shelf. Our results show that over the last 120,000 years, multi-year sea ice extent was greatest during Marine Isotope Stage (MIS) 2 and possibly during MIS 4, with more extended first-year sea ice during MIS 3 and MIS 5. Sea ice extent during the Holocene (MIS 1) has been less than at any time in the last 120,000 years.

Page 2, line 1 "reservoirs" Thanks.

Page 3, equations 2-3 and surrounding text. I appreciate that correcting for crustal Na in Greenland is not straightforward, but you need to explain what your rationale is here better, and when you do, it doesn't completely make sense. You are implicitly assuming that when the salt/acid reaction has created excess Cl, there is no terrestrial Na (eq 3); and that if there is a Cl/Na ratio less than 1.8 that is entirely because of terrestrial Na, and not at all because of the salt/acid reaction leading to removal of Cl from aerosol. I am not sure why either of those assumptions should be correct. Normally one would use a crustal element such as Al or perhaps Ca to put limits on the terrestrial

correction. In the absence of that you need to revisit your method and at minimum state its limitations and suggest how much it may affect your results.

We greatly thank the referee for rising the issue. We also agree in the fact that both processes (terrestrial sodium inputs and dechlorination processes) could act simultaneously to modify the Na/Cl ratios. By looking at their 1994 paper, Hansson et al. (1994) wrote "The mean ratio between Cl- and Na+ never exceeds the sea water ratio for any climatic stage, i.e. never indicating an excess of Cl-, which is a further indication that the deviation from the sea water ratio is due to an additional (crustal) source of Na+ and not a fractionation process in the atmosphere between the sea salt elements". Our measurements, show in fact that the Cl/Na do exceed 1.8.

Thus, sea salt sodium (ssNa) has been calculated in two additional ways: using calcium and magnesium, both measured by ICP-SFMS (discontinuously in the Holocene, continuously in the glacial section). The analysis is described in Section 3. We also dedicate a section in the Appendix in which we infer from present day studies on desert storms the ratio of Ca/Mg and Ca/Na in 'asian dust'. These numbers are used to calculate ssNa. As far as calcium is concerned, Renland calcium concentrations showed unexpectedly high values during the glacial. This has been previously shown also in the GRIP core, probably due to the appearance of gypsum/carbonated dust. We thus account for the high calcium levels to provide a calculation of ssNa based on calcium, which turns out very similar to ssNa calculated using magnesium. We therefore consider two ssNa curves and two Brenr curves: Brenr, Mg and Brenr, Cl, calculated respectively from chlorine and magnesium concentrations. We will not attach here this whole section since it consists of a number of pages, plots and tables. It will be provided in the modified version of the MS.

Page 3, line 15-20. Thank you for using the word "suggesting" here. However I am not sure what point you are making in line 19-20. Why would we expect Holocene-like ice extent at 120 kyr when we are already well into the glacial inception?

We agree. The sentence has been written more clearly.

Page 3, line 26-28. The modelling study of Rhodes et al (Rhodes, R. H. et al., GRL, 45(11), 5572-5580, doi:10.1029/2018gl077403) doesn't really support your interpretation, as in present conditions, it says that both sites are overwhelmingly seeing OW conditions (Rhodes was looking at the influence of sea salt aerosol but I imagine modelling Br would produce a similar result). Thanks for rising this point. Rhodes et al. (2018) modeled an annual SISS/OOSS (sea ice sea salt / open ocean sea salt) of 0.21 at NEEM (fSISS annually integrated, Table S2 and Fig. 1), while at Renland this ratio drops to 0.04, meaning that at NEEM the sea ice influence would be 5 times greater than that at Renland. The ratio of the bromine enrichment values at the two locations are roughly 3-4 (if we consider Holocene averages, Fig. 3 in our manuscript), pointing to a similar conclusion, at least if a relative sea ice vs open ocean contribution are considered. On an absolute scale, it appears from Rhodes et al. (2018) that NEEM is indeed more influenced by open ocean rather than sea ice, at least just considering sodium. We therefore modify the text to better indicate that the different Brenr values at the two locations might reflect the RELATIVE influence of sea ice, being 3 to 5 times higher at NEEM than Renland.

The difference between Holocene values for the two sites could also be partly related to the fact that the Br\_enr measure is not really ideal because it does not measure the amount of Br reaching the site but rather the ratio of Br/sea salt aerosol. NGRIP of course receives much less sea salt than Renland so just a small amount of gas phase Br can induce a Br\_enr >>1. This may not fully explain the difference between the sites but it is a factor. It's too late now because this measure is embedded in

the literature, but the use of Br\_excess ([Br-Br\_seawater] instead of [Br/Br\_seawater]) would have avoided this problem.

We probably agree, but we point out that a core1-core2 comparison between Br\_excess curves could be interfered by the difference in snow accumulation.

Page 4, line 8 "a number of" Thanks.

Page 6, last line "Our reconstruction suggests.." Done.

Figure 2. I appreciate that Kindler only goes to 10 kyr but it is unhelpful that we don't see a climate record from 10-0 kyr. Can you infill with an 18O record to at least indicate to the reader that Holocene temperatures remain warm.

We now used NGRIP  $\delta^{18}$ O in Fig. 4 and specified that in the caption of Fig. 6.

Fig 3 caption, Fig 6 caption: Tzedakis et al is not an appropriate reference for orbital parameters. Please cite original papers by either Berger or Laskar. Thanks.

Appendix B. I am wondering why you call this "sea salt aerosol source area" when what you are interested in is the source of the gas phase Br. The back trajectories are for air masses so certainly not specific to aerosol.

Both gas-phase (Br) and aerosols (Br,Na) contribute to Br\_enr, so both need to be considered, so this paragraph (now Sect. 2.3) is now named "*Atmospheric reanalysis: the source region of bromine and sodium for the RECAP ice core*".

Fig B1 caption line 2 "constraint" (but can you explain what you mean by this constraint). Only the trajectories that crossed the 900 hPa isosurface were selected and considered for the analysis. The term constraint has been removed.

Fig B1 caption. Sorry for my ignorance but I never heard of a nabla before, can't you call it an inverted triangle? We now use diamond symbols.

**References**:**

[revised manuscript text omitted]

**REVIEW - REFEREE#2**

The paper by Maffezzoli presents the first data on sea ice coverage of the North Atlantic and the Greenland, Iceland, Norwegian (GIN) Seas based on an ice core recently drilled on a coastal ice cap from East Greenland (Renland ice core). Following the previous approach by Spolaor et al. (2016) they use the Br enrichment above sea salt concentrations (Br\_enr) linked to halogen explosions occurring on seasonal sea ice surfaces. In principle, the data and argumentation of the paper are convincing although at some points not explained in detail enough. The paper represents an important contribution to the field that will be of interest for ice core specialists, marine geologists and modelers alike. In its current state, however, the paper still suffers from some language issues (see annotated pdf file attached), some structural deficits and some issues with the argumentations that I outline in my general comments below. I am convinced that these changes can be accommodated and recommend to accept the paper after major revisions.

We really thank Reviewer#2 for the time she/he took in making a constructive review. We now modified the paper structure by incorporating all the analyses in the whole Methods section of the main text (except for a study on 'asian' dust chemical composition in the Appendix). The deglaciation section has been further analyzed and includes two additional PIP records. The Brenr curves are now two (Brenr,Mg and Brenr,Cl), depending on how ssNa is calculated (we refer to the answers to Reviewer#1 on this topic). The Brenr-transformation has now been removed. We thank Reviewer#2 for finding the paper interesting and for spending time to produce the annotated pdf.

General comments

- Page 2 Line 17-23, "We compare . . . Belt and Muller, 2013)": This text should only come at the end of section 2

Done.

- Section 2, 2nd paragraph: In this paragraph it is stated that the origin of Renland sea salt is the North Atlantic and the text refers to Appendix B. This is a crucial piece of information and should not be hidden in an Appendix but included in the main text.

This analysis is now part of the Section 2.3 (main text): "*Atmospheric reanalysis: the source region of bromine and sodium for the RECAP ice core*".

Moreover some more information on this statement would be helpful for the reader:

a) sea salt aerosol has a pronounced seasonal cycle with a broad maximum in the winter half-year. This also holds for eastern coastal sites (Oyabu et al., Polar Science, 2016). Thus, the Renland sea salt record is mainly representative for the winter half-year. The back trajectory analyses should be done both for the winter and summer half-year separately and shown in two panels.

The atmospheric reanalysis has been done on a seasonal basis; the results are shown in the 4 panels (seasons) of Figure 1. We note, however, that although sea salts inputs have winter maxima, bromine explosions occur during springtime and so do Br\_enr values, as it has been (broadly) observed in shallow core records (Spolaor et al., 2014, Maffezzoli et al., 2017, Vallelonga et al., 2017).

Note also that due to the limited lifetime of sea salt aerosol, short trajectories are likely to bring more sea-salt aerosol to Renland compared to longer trajectories.

Thanks for the comment. A line has been added: "...although the ocean regions closer to Renland are expected to be more significant as per the observed chemical signature".

b) transport pathway (trajectories) is one side of the coin, the sea ice source is the other. An additional figure/panel showing the multiyear and first year sea ice distribution in spring (at its maximum) would be helpful to support the claim that the source regions identified in the trajectory study are covered by FYSI or OW. The National Snow and Ice Data Center provides this information (https://nsidc.org/data/nsidc-0611).

A sea ice age plot has been added (Figure. 2), showing a representative winter maximum (Feb 1987) and summer minimum (Sept 2012) during the satellite era.

- A method section is missing. Again, this is hidden in an Appendix A but should be part of the main text.

The Method section is now fully described in the main text (Sect. 2) and includes three subsections: - 2.1: The 2015 RECAP ice core

- 2.2: Experimental: determination of Br, Na, Cl, Ca and Mg by mass spectroscopy
- 2.3: Atmospheric reanalysis: the source region of bromine and sodium for the RECAP ice core

- Page 3 line 15-20, "The Br\_enr . . . Holocene value.": This text should be the beginning of section 3 These lines have been now moved to the "Results and Discussion" section (Sect. 4).

- Page 3 line 22-29. Here the NEEM Br\_enr record is mentioned and compared to Renland. In order to allow the reader to make this comparison, the NEEM data should be shown in one of the figures! Done (Fig. 4).

- Page 4 line 1: Here the expression "tipping point" is used. There is no clear definition of what a tipping point is, but in climate science it is usually used for a rapid regime shift (see IPCC). The change from MYSI to FYSI to OW, however, is likely a gradual process. Accordingly, I would suggest to avoid the expression tipping point in the manuscript. Along this line, I think section 3.2 (linearization of the Br\_enr record) and its application on the 120 kyr record in section 3.3 does not provide added value and in fact is misleading due to the gradual nature of the MYSI/FYSI/OW transition. Instead of trying to force this mathematically to a monotonous function, I would recommend to just use color bars in the figures underlying the records to indicate where MYSI, FYSI or OW dominate the Br\_enr record. Moreover, mean Greenland temperature may not be the only parameter determining the amount of FYSI present (see next comment).

We agree, the term tipping point has now been removed. We also removed the Brenr-transformation.

- The high resolution data presented in Figure 3 clearly show that the YD is the time period of largest Br\_enr (clear maximum) during termination I, thus the largest FYSI presence. In particular Br\_enr is clearly higher in the YD than in the OD period. Note that there is no similar Br\_enr maximum during the OD/BA transition as seen in the YD at the point when temperature during the OD/BA transition is crossing the same temperature as found in the YD. Either this point is just missed in the record (unlikely), or NGRIP temperature alone is not able to fully explain the observations. Here additional information could be used to elaborate on this issue. First of all, FYSI is strongly dependent on the seasonal temperature variation, this should be mentioned somewhere. Models suggest (Buizert et al., Science, 2014) that temperature seasonality during the YD and OD was much higher than during the BA. This could explain why the YD has higher FYSI than the BA. The difference between YD and OD sea ice conditions (Br\_enr levels) may potentially be explained by the overall much lower temperatures encountered during the OD than in the YD (Buizert et al., 2014), which may push the OD sea ice

regime toward more MYSI. This difference may be linked to the generally higher AMOC strength in the YD compared to the OD (McManus et al., Nature 2004).

- Page 4 line 29-33: Here the paper by Rasmussen is referred to, but I am not sure based on the text provided - whether it is referred to correctly and whether the statement made in the manuscript is correct. Rasmussen et al. (2016) claim that in the North Atlantic south of Iceland SST warming already starts during stadial conditions, while in the GIN seas the warming starts only with the Greenland DO onset, i.e., when sea ice rapidly declines. Rasmussen et al., do not explicitly discuss the YD/BA/OD transitions and in fact their record does not show a clear early warming during the reduced AMOC conditions of the YD and OD. Accordingly, to make this statement would require high resolution Br\_enr data for selected DO events from the Renland record, which are not available yet. I would suggest to remove this statement and also the reference to mean ocean temperatures, Antarctica and CO2.

Thanks for these comment. We will provide a single comment. This section (4.1) has now been analyzed in greater detail. We agree with the comments on the differences between YD/BA and OD, adding also that the availability freshwater from melting ice sheets during the deglaciation could have facilitated the formation of fresh sea ice surfaces. We agree on the comparison with Rasmussen, which has now been removed. We left however the sentence on the mean ocean temperatures, Antarctica and CO2 since we believe it could be a meaningful point for a broader perspective. Two additional PIP records from the Norwegian Sea has been added the Figure 6. The section of the deglaciation now reads:

[revised manuscript text omitted]

- Page 6, line 26. Here you refer to the GI numbers. These should be included in the figures Done.

- Include Fig. 1 in Fig. 2 Fig. 4 now shows all the measured records from RECAP.

- Add the NEEM record in Fig. 2 or provide a separate figure for the NEEM/Renland comparison. Done (Fig. 4).

- Add color bars for sea ice conditions underlying figure 2 As it is the first plot, no error bars are colored in Figure 4. The colors appear in Figure 7, where the two regimes are discussed.

- add color bars for sea ice conditions underlying figure 3 The two regimes are just indicated with two colored bands (Fig. 6).

- remove figure 5 Done.

- remove the transformed BR\_enr in figure 6, add color bars Done (Fig. 7).

- move Appendix A to a Method section Done, now Sections 2.1 and 2.2 (Methods).

- move Appendix B to section 2 and add more information as outlined above Done, now Section 2.3 (Methods).

Specific Comments - see annotated pdf We really thank the Reviewer for the annotated pdf.

In conclusion, the authors should clearly state that their hypothesis is not yet fully demonstrated and their conclusions are only hypothetical. Currently, other interpretations are possibles.

We thank Reviewer#3 for her/his comment. We agree on the general point that the uncertainties related to the Brenr have to be clearly stated. The manuscript introduction now focuses on Brenr as a potential indicator of past sea ice conditions (1. Introduction: Brenr as a potential indicator for past sea ice conditions). The uncertainties and the key aspects that require future studies and investigations are

discussed. This paragraph includes both aspects that Reviewer#3 and Reviewer#1 pointed out, as well as other few aspects. This section is structured into three general aspects related to Brenr: activation, transport and deposition/postdeposition. The reader can find this section attached in the answer file to Reviewer#1 (pp2,3).

**120,000 year record of sea ice in the North Atlantic ?**

Niccolò Maffezzoli1,2, Paul Vallelonga2, Ross Edwards3,4, Alfonso Saiz-Lopez5, Clara Turetta1,6, Helle Astrid Kjær2, Carlo Barbante1,6, Bo Vinther2, and Andrea Spolaor1,6 1Institute 
[revised manuscript text omitted]

(1)

$$\underline{ssNa} = \longleftrightarrow < 1.8 \tag{2}$$

 $\underline{ssNa} Na = \underline{Na} \longleftrightarrow \geq 1.8$ (3)

where Brand ssNa ssNa + nssNa

15

where Br, ssNa and nssNa are the bromineand, sea-salt and non-sea-salt sodium concentrations in the ice samples  $\frac{1.8}{1.8}$ 20 and respectively, and  $(Br/Na)_m = 0.0062$  are respectively the chlorine/sodium and bromine/sodium sea water mass ratios (Millero et al., 2008).

Since sodium has a potential crustal contribution is the bromine-to-sodium mass ratio in sea water (Millero et al., 2008), assumed constant in time and space. Since sodium concentrations in ice cores can be interfered by terrestrial inputs (nssNa, from sodium oxide Na2O), which can be significant especially during generally contribute up to  $\approx 10-50\%$  during glacial

[revised manuscript text omitted]

By using the sodium-to-magnesium ratios in asian dust  $(Na/Mg)_t = 1.4$ , MIS1) and during the coldest glacialphases, with 1.23 (Eq. 6) and in sea water  $(Mg/Na)_m = 0.12$  (Millero et al., 2008) (the subscript 'm' refers to 'marine'), we are able to calculate ssNa and ssMg (Eqs 8-9):

$$ssNa = \frac{Na - (Na/Mg)_{t} \cdot Mg}{1 - (Mg/Na)_{m} \cdot (Na/Mg)_{t}}$$
(8)

10
$$\operatorname{ssMg} = \frac{(\operatorname{Mg/Na})_{\mathrm{m}} \cdot Na - (\operatorname{Mg/Na})_{\mathrm{m}} \cdot (\operatorname{Na/Mg})_{\mathrm{t}} \cdot Mg}{1 - (\operatorname{Mg/Na})_{\mathrm{m}} \cdot (\operatorname{Na/Mg})_{\mathrm{t}}}$$
(9)

In the Holocene, on average ssNa $\approx$ 0.99Na, while ssMg $\approx$ 0.50-0.90Mg. During MIS2, ssNa $\approx$ 0.6–0.7Na and ssMg $\approx$ 0.20–0.30Mg, while during MIS4 ssNa $\approx$ 0.50–0.60Na and ssMg $\approx$ 0.10–0.20Mg. In those Holocene samples where magnesium was not measured, we assume Na = ssNa, making on average a 1% error.

- The same procedure using calcium has been used to calculate ssNa from Antarctic ice cores (Röthlisberger et al., 2002),
  by solving the equations similar to Eq. 8 and Eq. 9 with Mg replaced by Ca. In the RECAP record, such correction (with asian dust (Na/Ca)t=0.38 and (Ca/Na)m = 0.038 (Millero et al., 2008) unrealistically predicts ssNa ≈ 0 throughout the glacial period. Small changes in the (Na/Ca)t value do not significantly change this result. It appears that higher-than expected calcium is deposited at Renland, and therefore a lower (Na/Ca)t is to be used. Excess of calcium during the glacial period, has been reported in Greenlandic ice cores, in the form of calcite (CaCO3), dolomite (CaMg(CO3)2) and
- 20 gypsum (CaSO4 · 2H2O) (Mayewski et al., 1994; Maggi, 1997; De Angelis et al., 1997). In their investigation of the GRIP core Legrand and Mayewski 1997 find larger increase in sufate ( $SO_4^{2-}$ ) in Greenland compared to Antarctica during glacial times. Such large enhancements of the sulfate level are well correlated with calcium increases, but not with MSA, suggesting that sulfate levels are related to nonbiogenic sulfur sources (gypsum emissions from deserts, for instance) (Legrand and Mayewski, 1997). Different calcium sources were likely active during the glacial as compared to warmer periods, but it is unclear whether these
- 25 were new active continental sources or continental shelfs which became exposed due to a lowered sea level. Similarly to what has been observed in other Greenland ice cores, extra calcium is also found in the RECAP core during the glacialminima at ~, as illustrated from the scatter plot with magnesium (Fig. 3). From the calcium-magnesium relation, we estimate that during the coldest and most arid glacial times (Ca/Mg)t = 23kyr (±1 (blue fit in Fig. 3), while in other periods we assume that the composition remained the same as the modern Asian dust composition (Appendix A):

$$\quad \left(\frac{Ca}{Mg}\right)_{t,modern} = 3.7 \tag{10}$$

This hypothesis is substantiated by the value found for the lower envelope for the Holocene, interstadial late MIS5 samples (green points in Fig. 3) as well as for composition of the bottom 22 meters of the core (562–584 m, red points in Fig. 3):  $(Ca/Mg)_t = 3.23 \pm 0.2(1 \text{ (red fit)})$ . We note that these bottom measurements do not appear in any time series since the core chronology ends at 120 kyr (562 m), although we here suggest that the RECAP bottom 22 meters may date back to the

5 previous interglacial, the Eemian.

Sea-salt sodium concentrations (ssNa) can therefore be calculated using calcium, by replacing in Eq. 8 Mg $\rightarrow$ Ca and by using (Ca/Na)m = 0.038 and (Na/Ca)t calculated by:

$$\left(\frac{Na}{Ca}\right)_{t} = \frac{(Na/Mg)_{t,modern}}{(Ca/Mg)_{t,glacialfit}} = \frac{1.23}{23} = 0.054$$
(11)

This latter value is not too far to the value (0.036) empirically found by De Angelis et al. 1997, who calculated ssNa in the
GRIP core based on a mixed chlorine-calcium method. The calculation of RECAP nssCa reveals that calcium contains almost a purely crustal signature: nssCa ≥ 0.95Ca throughout the record.

The ssNa curves calculated with magnesium and calcium (ssNaMg, ssNaCa) differ by less than  $1\sigma$ ), while appreciable differences between these two curves and the chlorine one (ssNaCl) are only found during MIS2 ) and at  $\sim$ 72 kyr (MIS4 (Fig. 4b). Since ssNaMg  $\simeq$  ssNaCa, for the following discussion we will only consider the two Brenr = 2.6±0.4(series based on

- 15 the chlorine and magnesium correction:  $Br_{enr,Cl}$  and  $Br_{enr,Mg}$  (Fig. 4g). Standard error propagation is carried out to yield the final  $Br_{enr}$  uncertainties. This analysis along with past studies demonstrate that the sea-salt and non-sea-salt calculations of elemental concentrations depend on the chemical composition of the dust that is deposited at the ice core site. The dust composition has a spatial variability which should be considered, while the use of tabulated reference values should be discouraged.
- From the calculated RECAP  $Br_{enr}$  curves, we now investigate past sea ice conditions in the 50-85 °N North Atlantic Ocean, based on the aforementioned hypotheses on the  $Br_{enr}$  use as an indicator of first-year sea ice conditions. Because of the RECAP location, the record is sensitive to ocean processes and sea ice dynamics (Cuevas et al., 2018). We compare our sea ice record with PIP25 results from three marine sediment cores drilled within the Renland source area: the Fram Strait, the Norwegian Sea and the North Icelandic shelf. The PIP25 index is a semi-quantitative indicator of the local sea ice condition
- 25 at the marine core location. It is calculated by coupling the sediment concentration of  $IP_{25}$ , a biomarker produced by diatoms living in seasonal sea ice, with an open water phytoplankton biomarker (brassicasterol or dinosterol, hence  $P_BIP_{25}$  or  $P_DIP_{25}$ ). Briefly, the PIP25 index is a dimensionless number varying from 0 to 1), MIS4). At the Eemian termination: PIP25  $\approx 1$  indicates perennial sea ice cover; PIP25  $\approx 0$  indicates open water conditions, while intermediate PIP25 
[revised manuscript text omitted]
{array}{l} \mathrm{Br}_{\mathrm{enr}} \longrightarrow \mathrm{Br}_{\mathrm{enr}} \\ T > \mathrm{T}_{\mathrm{NGRIP}}(12.4 - 11.8 \mathrm{kyr}) = -44.6 \pm 0.9 (2\sigma)^{\circ} \mathrm{C} \end{array}$

$$\begin{split} \mathbf{Br}_{\mathrm{enr}} &\longrightarrow \overline{\mathbf{Br}_{\mathrm{enr}}} + \left(\overline{\mathbf{Br}_{\mathrm{enr}}} - \mathbf{Br}_{\mathrm{enr}}\right) \\ T &< \mathbf{T}_{\mathrm{NGRIP}}(12.4 - 11.8 \mathrm{kyr}) = -44.6 \pm 0.9 (2\sigma)^{\circ} \mathrm{C} \end{split}$$

during the Older Dryas compared to the Bølling-Allerød and the Younger Dryas may be linked to the overall much lower temperatures during this period (Buizert et al., 2014), higher surface water salinity due to less freshwater inputs from melting

15 ice sheets and a generally weaker AMOC. Standard error propagation is carried out when transforming in Eq. ??.

The transformed-variable

**4.2 The 120,000 year $Br_{enr}$ record**

We now apply the previously mentioned temperature-based discrimination of the two sea ice regimes to the 120,000 year
 Brenr,Mg record (Fig. ??). The same analysis performed on the Brenr,Cl curve can be found in the Appendix (Fig. ??, lower panels) is now to be interpreted as being linearly variable between OW and MYSI conditions, therefore providing an absolute index of sea ice extent within the Renland source area.

??). The regime type in each ice sample is represented by the color of the error bars: blue for the FYSI/MYSI regime and red for the FYSI/OW regime. In order to test the sensitivity of the regime output on the threshold value, 4 scenarios are

computed, using a  $\pm 1\sigma$  and a  $\pm 2\sigma$  value around the temperature threshold mean value  $\div \overline{T} = -44.6$  °C (Fig. ??. Left: -2:top; +??). Except 2 :bottom. Right: -1:top; +(1) samples at 20 kyr showing a different regime type during the deglaciation (in the  $\overline{T}$ -2(1) $\sigma$  :bottom). We note that for the simple 2-state model applied here, discontinuities could occur for adjacent samples integrating temperature values close to the temperature threshold. Only at a few time periods, however, is scenario- top panels in Fig. ??), the regime discrimination significantly affected by the chosen threshold value (see them discussed mentioned in the following section). MIS1 and MIS5 are characterized by the 'warm' discussion is invariant with respect to the 4 scenarios. MIS 5 is characterized by increasing Brenr.Mg values in the FYSI/OW regime , while the 'cold' from 120 to 80 kyr ago

5 (Greenland Interstadial, GI-21). We interpret this trend with increasing FYSI extent in the North Atlantic. From the end of GI-21 the Brenr regime changes to FYSI/MYSI regime occurs during MIS2 and MIS4. MIS3 shows a mixture between the two regimes.

**4.3 The 120,000 year sea ice record**

10

The two North Atlantic sea ice time series, calculated by applying the as the NGRIP temperature drops during late-MIS 5 and MIS 4. As compared to the levels reached during GI-21 (8±2transformation (i.e. allowing for the biggest discrepancies), are

shown in Fig. ??. It is worth noting that since the sample integrates 1500 years, these two records have been downscaled to 1500 years.

Some discrepancies between the two reconstructed sea ice records are found. The first one is at the 23.3-17.3 kyr (Last Glacial Maximum), with reduced sea ice predicted for the 'lower' temperature threshold (see also Fig. ??). The second one,

- 15 at 35.3-33.8 kyr (integrating Greenland Stadial GS-7, and Greenland Interstadial GI-7), results in opposite sea ice scenarios. Finally, at 72.8-71.3 kyr (integrating GI-19.2-1), lower Brenr,Mg values are found during GS-21, GS-20 and GS-20), both scenarios indicate a decrease in sea ice but of slightly different magnitude. The output of the transformation is sensitive to the chosen temperature threshold, and therefore could result in anomalous sea ice states during those climate periods when the temperature was close to the threshold. If the GS-19 (5±2) and on average during MIS 4 (7±1transformations are considered

[revised manuscript text omitted]

---

## Referee Report (RR1)

In this paper by N. Maffezzoli and coauthors a new record of Bromine concentration measured along the Renland ice core is presented. As already observed by the other reviewers, the authors consider Br enr as a proxy of FYSI, but this is still to be proved. With respect to the original version of the paper, the revised one clarify this point and the "certain assertions" have been changed and transformed in "probable conclusions". I appreciate this change all along the manuscript but, since Bromine is thought to be a marker just of the FYSI, I would suggest to change the title in "120,000 year record of First-Year Sea Ice in the North Atlantic ?".

The paper is well presented and written and the discussed dataset seems to be robust in terms of accuracy and precision, given the good inter-laboratory exercise here described.

I have two major concerns about the use of Br enr as a proxy to reconstruct sea ice (or FYSI):

1) Bromine in the snow and ice has been deposited through dry and (mainly, in this case) wet depositions. How stable is this element once is deposited in the snow? Assuming that bromine concentration is really dependent by FYSI only, is the recorded climatic signal well preserved in time? This point is just briefly discussed in terms of "possible photolytic reemission of bromine", but this phenomenon is strongly dependent on the accumulation rate and I expect a high year to year accumulation variability in the RECAP core, due to its proximity to the ocean. Furthermore, this problem can be amplified when looking at larger time scales (glacial/interglacial) as done in this paper. I think that this point should be described and discussed in more details.

2) The calculation of Br enr could represent a caveat of the method. The "correction" of the Br concentration using Na+ concentration could produce an artefact in the Br enr profile: the higher the Na+, the lower the Br enr. Being Br part of SSA, we can observe also in this paper an agreement (not always good) between ssNa and Br profiles. To better clarify this point: in the last years ssNa+ has been used as a potential proxy of sea-ice extent (Iizuka et al., 2008, JGR, Severi et al., 2017, Chemosphere) and its use to calculate an enrichment factor for Bromine, could transfer the information hidden in the Na+ profile to the Br enr one. The proof of this is clearly visible comparing the ssNa profile with the Br enr one (they are clearly anti-correlated, see figure below). I think that both markers are reflecting changes in the sea ice conditions and a correlation between the two would help understanding this point. If the correlation between the two is good enough I think that Br concentration (or flux) itself would be a reliable marker without any correction or enrichment factor calculation. For example, in the paper by the same authors' team using Iodine

(Corella et al., 2019), they did not make any correction and used directly the I concentration. Also this aspect should be discussed along the main text and a correlation plot between ssNa and Br should be added (in the SI would be fine as well).

Minor changes:
Page 8 line 7: the values 0.36 and 0.54 are not so close as the authors say (it's 50% higher!). Thus, change this sentence.

Simonsen et al, Nat.Comm is now published and should be added to the references list.

---

## Author Response (AR2)

**List of relevant changes**

- **The discussion on bromine remobilization has been included (Appendix A).**
- **Br-ssNa correlation plot has been added (Appendix C).**
- **Figure 7 is now presented in just one scenario (Temperature T). The temperature sensitivity tests (T±1σ, T±2σ) are presented in the Appendix D (Figs D1, using $Br_{enr,Mg}$ and D2, using $Br_{enr,Cl}$).**

**As per the ice core age model, the Simonsen 2019 paper has been referred.**

**Referee #1**

We thank Referee#1 once again for taking the time to read the manuscript after such a long time. We appreciated her/his positive view of the revised version.

The authors have really done a very major revision of this paper and have acted on all the main comments made by the reviewers including me. In particular they have:
• Made it much clearer that their interpretation of Br_enr is not yet confirmed
• Done a complete revision of the calculation of the nss components with multiple options to show the sensitivity to different methods
• Revised the way they present the FYSI/OW and FYSI/MYSI discussion, removing the previous transformation

I cannot say I agree with every last sentence of their interpretation now, but it is a very much sounder scientific paper that presents the data but allows the reader to assess the viability of the interpretation. I have some minor suggestions (and even more minor typos) for the authors to consider. However overall, I thank them for being open to so many changes to their presentation and believe the paper is now almost ready for publication.

Abstract: I appreciate the caveats added here. I would suggest two extra changes that I think will better represent the way the findings are now presented.
line 8 Instead of "We find that", I propose "Our interpretation implies that"
Done.
line 11 "as supported by sea ice records…". I think it might be more helpful to instead say "Our interpretation is consistent with sea ice records…"
We modified to "... started to dominate, consistent with sea ice records from the eastern Nordic seas and the North Icelandic shelf."

Page 2, Line 33. Spolaor didn't really show opposite results, rather they tried to explain their data by assuming the opposite in their model. Perhaps better "a model run by Spolaor assumed the opposite".
Correct, we agree. We further elaborated on this topic, including the study by Li et al. (2014). The part related on the transport riddle now reads:
*"… It appears that fine aerosol are enriched in bromine, while coarse particles are depleted (Legrand et al. (2016), Koenig pers. comm.). Additionally, the two phases likely have different atmospheric residence time, thus the final bromine-to-sodium ratio found in the snow might be function of the transport duration. In general, bromine-reactive surfaces would increase the atmospheric residence time of bromine species, enhancing the bromine-to-sodium ratios away from the sea ice or ocean sources. On this topic, from springtime Arctic snow samples collected from a transect directed inland, Simpson et al. (2005) showed that sodium is deposited faster than bromine, suggesting the role of longer lasting gas-phase bromine (Fig. 3 in their study). From samples collected during a coast-to-inland Antarctic transect (Zhongshan Station--Dome A), Li et al. (2014) showed much reduced spatial gradients between sodium and bromine snow concentrations, if compared to Simpson and co-workers (Fig. 2 in their study). In a similar inland-directed Antarctic transect (Talos Dome--Dome C), Spolaor et al. (2013) presented simultaneous bromine and sodium deposition fluxes from 3 sites and a model run aimed at explaining the experimental results (Fig. 6 in their study). The authors note that for their results to be explained by the model, the deposition velocity of HBr needs to be set at least 3 times larger than the average sea-salt aerosol deposition velocity. To conclude, to which extent transport processes can interfere with a sea ice source effect in either the Arctic or Antarctica is to be further investigated."*

Page 3, line 4, you mean "shed light" not "shade light".
Thanks.
Page 4, line 25 "1.1 ppb (Na)"
Thanks.
Page 7, line 7 Asian should have capital A
Thanks.
line 21 sulfate not sufate
Thanks.
Page 8, line 27 "for the" not "for the for"
Thanks.

Fig 7. It's really hard to see the details of the Br_enr plots, including the colour changes. I would suggest that this figure presents as a large plot just one version (logically with Tbar) and then shows the 4 alternatives in the supplement, perhaps as another figure in appendix B. This would be clearer. Obviously it would also require changes in the text to reference the different parts of the figure.
Done. We have also included the Fram Strait PIP record from Muller and Stein.

Figure 6. I notice that Muller and Stein have some intermediate PIP values again at 24 ka which don't seem to be represented by an increase in your Br data and which are just off the edge of this figure. Worth a comment?
The reviewer is correct. The Fram Strait sea ice record indicates variable seasonal sea ice (intermediate/high PIP values) from 26 to 19 kyr BP. The sea ice record in the Norwegian Sea also indicates lower values prior to the ca. 22 kyr maximum.  When discussing MIS2, at the end of Section 4.2, the text has been modified by considering the 2 marine records:

"...MIS 2 is characterized by a FYSI/MYSI regime and progressively decreasing $Br_{enr,Mg}$ values (Fig. 7e), with a minimum reached ∼23 kyr ago, during the Last Glacial Maximum: $Br_{enr,Mg}$ (LGM)=3.3±0.3. We interpret this negative $Br_{enr,Mg}$ (and $Br_{enr,Cl}$ , Fig. D2) trend with progressively increasing MYSI conditions in the whole North Atlantic, that reached a maximum during the LGM. Relatively high and increasing $PIP_{25}$ values (≈0.5-1) are found in the Norwegian Sea record (Fig. 7d), with maximum sea ice found registered ca. 20-23 kyr BP. In broad agreement with the Norwegian record is the sea ice record from Fram Strait Fig. (7c), the latter showing high (≈0.5-1) and variable PIP 25 values until the longer lasting maximum ca. 19 kyr BP. Overall, the cold MIS2 period is characterized by decreasing seasonal sea ice and increasing multiyear sea ice with a maximum reached ca. 20 kyr BP."

Additional references:
Li, C., Kang, S., Shi, G., Huang, J., Ding, M., Zhang, Q., Zhang, L., Guo, J., Xiao, C., Hou, S. and Sun, B., 2014. Spatial and temporal variations of total mercury in Antarctic snow along the transect from Zhongshan Station to Dome A. *Tellus B: Chemical and Physical Meteorology*, *66*(1), p.25152.

**Referee #4**

In this paper by N. Maffezzoli and coauthors a new record of Bromine concentration measured along the Renland ice core is presented. As already observed by the other reviewers, the authors consider Br enr as a proxy of FYSI, but this is still to be proved. With respect to the original version of the paper, the revised one clarify this point and the "certain assertions" have been changed and transformed in "probable conclusions". I appreciate this change all along the manuscript but, since Bromine is thought to be a marker just of the FYSI, I would suggest to change the title in "120,000 year record of First-Year Sea Ice in the North Atlantic ?".

Since the discussion is not only focused on first year sea ice but on sea ice in general, and it also includes sea ice records from marine cores, we believe that the title can be appropriate.

The paper is well presented and written and the discussed dataset seems to be robust in terms of accuracy and precision, given the good inter-laboratory exercise here described.

We thank Referee#4 for taking her/his time spent reading this version, the previous version and the previous comments.

I have two major concerns about the use of Br enr as a proxy to reconstruct sea ice (or FYSI):
1) Bromine in the snow and ice has been deposited through dry and (mainly, in this case) wet depositions. How stable is this element once is deposited in the snow? Assuming that bromine concentration is really dependent by FYSI only, is the recorded climatic signal well preserved in time? This point is just briefly discussed in terms of "possible photolytic reemission of bromine", but this phenomenon is strongly dependent on the accumulation rate and I expect a high year to year accumulation variability in the RECAP core, due to its proximity to the ocean. Furthermore, this problem can be amplified when looking at larger time scales (glacial/interglacial) as done in this paper. I think that this point should be described and discussed in more details.

We agree with the reviewer that halogen reemission is a major point of discussion. We add this discussion to the appendix since it requires quite some space so that the flow is not altered.

As a general comment, we are not sure that bromine is mainly wet deposited at Renland (any attempt to calculate dry/wet contribution would be extremely qualitative primarily due to uncertainties affecting snow scavenging coefficients of bromine species) . The wet/dry contributions likely change throughout the last 120,000 years and if reemission occurs, the absolute mass of bromine loss from the ice cap would change in turn throughout time. We summarize here the experimental investigations that have been carried out on this topic ever since the study by Dibb et al. (2010) in Antarctica and in the Arctic.

Added text (Appendix):

---------Antarctica--------
**McConnell et al. (2017)** investigated simultaneous sodium and bromine fluxes across Antarctica (see their figure S4 in the appendix showing the reconstructed fluxes from an array of 11 cores from McConnell et al. (2014)). In particular, the J(Na) and J(Br) - Accumulation rate (A) linear relations (impressive J(Br)-A linear fit from 50 to 400 kg m$^{-2}$ yr$^{-1}$) suggest that, while sodium loss is negligible, bromine reemission is quantified to 17±2 µg m$^{-2}$ yr$^{-1}$ (negative intercept, i.e. snow-air flux). According to the results, the bromine loss from Antarctic snowpack would be (I point out that this exercise works on the hypothesis the the bromine air concentration is all these sites, but the goodness of their fit would suggest that this is the case): 65% at sites with A=50 kg m$^{-2}$ yr$^{-1}$, 32% at sites with A=100 kg m$^{-2}$ yr$^{-1}$, 22% at sites with A=150 kg m$^{-2}$ yr$^{-1}$, 16% at sites with A=200 kg m$^{-2}$ yr$^{-1}$, 13% at sites with A=250 kg m$^{-2}$ yr$^{-1}$, 11% at sites with A=300 kg m$^{-2}$ yr$^{-1}$. These negative fluxes would be representative of Antarctic conditions for the duration of the records, i.e. few centuries to max 2 millennia (see Supp. Table 2 in the 2014 ref). In their Dome C (A=30 kg m$^{-2}$ yr$^{-1}$) study, **Legrand et al. (2016)** analyzed bromine

content in a 110cm 2-cm resolution snowpit. Their measurements show that the upper 2 cm contain 1.25 times more bromine that the 2-4 cm layer, 2.5 times more than the 4-8 cm layer and ca. 10 times more than the 8-12 cm layer. Based on the higher bromine concentrations towards the surface the authors suggest a possible remobilization from the snow. In testing the hypothesis, the authors model whether the measured bromine snow content is high enough to sustain the 1.7 pptv air concentration of $Br_y$ they measured in the overlying atmosphere. They conclude that to explain such value, the snow bromine storage should be 35 times larger than the actual measurements. They conclude that in Dome C their investigation does not support the importance of snowpack bromine emissions. This conclusion contrasts with what was concluded by McConnell and co-workers.

From surface snow experiments carried out in Dome C in 2014 (Antarctica, A=30 kg m$^{-2}$ yr$^{-1}$) **by Spolaor et al. (2018)**, the authors observe constant bromine concentrations from mid December to mid January. In a similar experiment carried out the following year, from late November to late December 2015, higher bromine concentrations were found in November with a decreasing trend towards the end of the year. The authors relate this drop to a change in the airmass transport, coastal followed by inland pathways, as inferred from back trajectory modeling, rather than to bromine loss from the snow.

From another surface snow experiment carried out at GV7 (Antarctica, A=270 kg m$^{-2}$ yr$^{-1}$), **Vallelonga et al. (QSR, in review)** compare bromine fresh surface snow concentrations with snowpit values, and suggest, on the basis of similar values between the values in the surface samples and in the surface of the snowpit (that would integrate the values of the surface samples), that at this site bromine is stable after deposition (whereas McConnell et al.'s calculation would predict a loss of 12%). Looking at the 100-300 µg m$^{-2}$ yr$^{-1}$ annual bromine (2010, 2011, 2012, 2013) fluxes calculated along the Talos Dome – GV7 traverse (**Maffezzoli et al., 2017**), a negative flux of 17±2 µg m$^{-2}$ yr$^{-1}$ would imply that, if present, the bromine loss can be quantified in this region as 6-17 %. As McConnell and co-workers point out, it is surprising that the snow-air bromine flux they found is not dependent on snow accumulation (their fit appears linear until the last point, 400 kg m$^{-2}$ yr$^{-1}$).

----Arctic-----
In the Arctic, the most robust piece of evidence results from hourly resolved surface snow measurements carried out in coastal Svalbard, showing insignificant bromine photolytic loss across night-day cycles **(Spolaor et al. 2019, accepted ACP, see Fig. 6)**.

In an experiment described in **Vallelonga and coworkers (QSR, in review)** carried out in Ny-Alesund (Svalbard, A=600 kg m$^{-2}$ yr$^{-1}$), sodium and bromine were measured in surface snow daily from April to June (spring to summer). From April to late May (afterwards surface melting and positive temperatures led to 80% sodium drainage from the snowpack) Brenr values decreased by a factor 2 (max). It is difficult however, to distinguish whether this decrease is due to bromine reemission or to early spring bromine explosions from sea ice, located ca. 150 km away from the sampling site, enriching the spring deposited snow layers.

To conclude, field experiments aimed at quantifying bromine loss from the snowpack are made difficult by the concurrent effect of other variables (source effects, changes in transport pathways, other post depositional processes) that can act simultaneously. Estimates of bromine loss from Dome C (Antarctica) have led to not unique conclusions. This site is likely one of the most challenging to carry out this study since the snow accumulation is so limited that it is challenging to decipher whether any loss can be attributed to reemission or wind erosion processes, exposing older layers having generally different bromine concentrations. In other parts of Antarctica, McConnell et al. (2017) suggest bromine snow-air flux of loss of 17±2 µg m$^{-2}$ yr$^{-1}$ regardless of snow accumulation, while at GV7 bromine loss

was not detected (Vallelonga et al., QSR, in review). In the Arctic, photolytic loss was not detected in Svalbard. Other experiments have been shown inconclusive due to the presence of surface melting. At sites like Renland (where Holocene melting occurs, see Taranczewski, in discussion 2019), the extremely high snow accumulation could in general suggest reduced bromine loss, but again, the fivefold decrease snow accumulation during the glacial (preliminary $A_{glacial, stadials}$ = 100 kg m$^{-2}$ yr$^{-1}$ (*Vinther unpublished*)) may introduce a possible bromine loss during periods of reduced accumulation, thus increasing $Br_{enr}$ values during the coldest parts of the record. To test these hypotheses and especially the bromine loss during the glacial, a surface study should be carried out at a 100 kg m$^{-2}$ yr$^{-1}$ accumulation site in Greenland. To conclude, at present there is not enough consistent data available to quantify the possible bromine loss from the Renland ice cap but, if present and dependent on accumulation, the effect would be to increase the measured $Br_{enr}$ values during the coldest sections of the glacial period.

2) The calculation of Br enr could represent a caveat of the method. The "correction" of the Br concentration using Na+ concentration could produce an artefact in the Br enr profile: the higher the Na+, the lower the Br enr. Being Br part of SSA, we can observe also in this paper an agreement (not always good) between ssNa and Br profiles. To better clarify this point: in the last years ssNa+ has been used as a potential proxy of sea-ice extent (Iizuka et al., 2008, JGR, Severi et al., 2017, Chemosphere) and its use to calculate an enrichment factor for Bromine, could transfer the information hidden in the Na+ profile to the Br enr one. The proof of this is clearly visible comparing the ssNa profile with the Br enr one (they are clearly anti-correlated, see figure below). I think that both markers are reflecting changes in the sea ice conditions and a correlation between the two would help understanding this point. If the correlation between the two is good enough I think that Br concentration (or flux) itself would be a reliable marker without any correction or enrichment factor calculation. For example, in the paper by the same authors' team using Iodine(Corella et al., 2019), they did not make any correction and used directly the I concentration. Also this aspect should be discussed along the main text and a correlation plot between ssNa and Br should be added (in the SI would be fine as well).

We agree that ssNa and Br have a common source (open water + sea ice). Their correlation in glacial ice is positive but weak $\rho_{glacial}$= 0.32.
"..its use to calculate an enrichment factor for Bromine, could transfer the information hidden in the Na+ profile to the Br enr one ." According to the Brenr hypothesis, if significant ssNa input is sourced from sea ice (likely from SSA from blowing snow), yielding high ssNa ice concentrations, concomitant high bromine from SSA would induce bromine explosions, thus high Brenr ice values as well, if extended FYSI surfaces are present.
The problems we see in using the absolute ice concentrations of the 2 impurities alone are:
1) effects of open water source (and/or, as suggested some ca. 20 years ago, interferences by storm strength changes)
2) the effect of accumulation rate changes (Eq. 1 Schupbach 2018) . This effect is clearly visible in the fact that when the accumulation rate is rather constant (i.e. Holocene), the ssNa/Br correlation decreases ($\rho_{Holocene}$= 0.1, unpublished).
We believe that by taking the ratio between Br and ssNa concentrations the dependence on accumulation changes and other sources is lowered. The (negative) correlation between ssNa and Brenr is significant by construction, therefore it does not add information.

Ideally, one would calculate Br_air and Na_air from the respective ice concentrations and dividing them, attempting the simple approach in Fisher (2015) or Schupbach (2018). This exercise, however, is

difficult since snow scavenging factors of the two species are known but with large uncertainty. The exercise is easier if the accumulation does not change much compared to the elemental concentration variability. It is the case of Corella et al. (2019), where the Holocene iodine concentration variability was interpreted as variability in iodine atmospheric concentrations since during the last 11 kyr the accumulation changes have been much more reduced compared to the observed ice iodine concentration changes.

Overall, we still think that it is worth showing the $\rho(Br,Na)$ to suggest a common source. The plot has been added in the Appendix and the following text has been added:
"A weak positive correlation ($\rho=0.32$, $p<10^4$, $n=144$) is found between Br and $ssNa_{Mg}$ concentrations in glacial ice (Fig. C1, Appendix C) suggesting that although the two elements share a common marine source, other source/transport effects are playing a role in driving changes in their respective ice concentrations."

Minor changes:
Page 8 line 7: the values 0.36 and 0.54 are not so close as the authors say (it's 50% higher!). Thus, change this sentence.
The sentence has been modified to:
"As a comparison, in the GRIP core, where the ssNa calculations were based on a mixed chlorine-calcium method, De Angelis et al. 1997 found empirically that $(Na/Ca)_{t,glacial} = 0.036$ in glacial ice (significant correlation, $\rho=0.7$) and $(Na/Ca)_{t,modern} = 0.07$ (weak correlation, $\rho=0.2$) in Holocene ice.".

Simonsen et al, Nat.Comm is now published and should be added to the references list.
Done.

Additional references:
- McConnell, J.R., Burke, A., Dunbar, N.W., Köhler, P., Thomas, J.L., Arienzo, M.M., Chellman, N.J., Maselli, O.J., Sigl, M., Adkins, J.F. and Baggenstos, D., 2017. Synchronous volcanic eruptions and abrupt climate change∼ 17.7 ka plausibly linked by stratospheric ozone depletion. *Proceedings of the National Academy of Sciences*, *114*(38), pp.10035-10040.

And in particular, the supplementary information from this link:
https://www.pnas.org/content/pnas/suppl/2017/08/29/1705595114.DCSupplemental/pnas.1705595114.sapp.pdf

- Taranczewski, T., Freitag, J., Eisen, O., Vinther, B., Wahl, S. and Kipfstuhl, S., 10,000 years of melt history of the 2015 Renland ice core, East Greenland., *The Cryosphere, in discussion (2019).*

- McConnell, J.R., Maselli, O.J., Sigl, M., Vallelonga, P., Neumann, T., Anschütz, H., Bales, R.C., Curran, M.A., Das, S.B., Edwards, R. and Kipfstuhl, S., 2014. Antarctic-wide array of high-resolution ice core records reveals pervasive lead pollution began in 1889 and persists today. *Scientific Reports*, *4*, p.5848.

- Spolaor, A., Barbaro, E., Cappelletti, D., Turetta, C., Mazzola, M., Giardi, F., Björkman, M. P., Lucchetta, F., Dallo, F., Pfaffhuber, K. A., Angot, H., Dommergue, A., Maturilli, M., Saiz-

Lopez, A., Barbante, C., and Cairns, W. R. L.: Diurnal cycle of iodine and mercury concentrations in Svalbard surface snow, Atmos. Chem. Phys. Discuss., https://doi.org/10.5194/acp-2019-285, in review, 2019.

- Spolaor, A., Angot, H., Roman, M., Dommergue, A., Scarchilli, C., Vardè, M., Del Guasta, M., Pedeli, X., Varin, C., Sprovieri, F. and Magand, O., 2018. Feedback mechanisms between snow and atmospheric mercury: Results and observations from field campaigns on the Antarctic plateau. *Chemosphere, 197,* pp.306-317.

- Vallelonga, P., Maffezzoli N., Saiz-Lopez A., Spolaor A., Sea ice reconstructions from halogens in ice cores, *Quaternary Science Reviews,* in review, 2019.

**Referee #5**

Building on previous efforts by the same group, this paper adds another interesting element to the possibility to use Br enrichment as a proxy for the extent of FYSI. Initial reviews were critical yet highly constructive and the authors have reacted positively and humbly, making major changes to their paper and their efforts are laudable.
We thank Referee#5 for taking time to read the MS versions and the previous review. We also thank her/him for this comment.

The current revised version reads well, does explain the limits and caveats of the approach and I have to say almost adds a convincing touch to the hypothesis. Thus, FYSI extent variations could indeed be determined from Br enrichment, and lower values would be indicative of either MYSI or OW (open water).

Given that the hypothesis is still being tested, numberless details could be discussed further. However, my main impression is that the paper is interesting, for the most part the interpretations are logical and the data and discussion represent a useful contribution to the topic, so that I am happy to recommend publication of the paper in CP.

I however have a question that disturbs me a bit, and I am not sure whether I or the authors missed something. From the data of Figure 6, the authors used the NGRIP temperature during the YD as the threshold for a change in regimes between MYSI/FYSI and FYSI/OW. But that same temperature was prevalent during 16-18 kyr, just before the OD, and no change in regime took place then. According to the authors, during the YD, temperature seasonality and fresh water input were different, which explains why, despite the cold temperature, a maximum in FYSI was reached. But then, since obviously temperature is not the only intervening variable, and since during the YD conditions were arguably particular and not representative of what took place during glacial epochs, why apply this threshold to the whole 120 kyr record? I guess I will not be the only reader to wonder, and some explanation is in order. By the way, why is the one low Br data point around 12.5 kyr never discussed? There are just 3 points with a high Br value during the YD, so one point, i.e. 25% of the data during or near the YD, certainly cannot be ignored and the difficulty swept under the rug. Likewise, is the high Br point at 13.8 kyr meaningful or just noise in the data? Perhaps some more detailed investigation would yield more clues about the interactions between the Br signal and climate variables.

- We agree with Reviewer5. The particular conditions during the YD have been invoked (following the suggestions of Reviewer#2 during the first review round) as possible reasons to explain the relative Brenr levels between the YD/BA and OD periods. Such conditions were probably not representative for the entirety of the glacial period (at least in general, for e.g. temperature seasonality, globabl ice sheet volume or meltwater inputs), therefore we believe that temperature (although NGRIP air temperature) could still be used to differential between the suggested regimes and hence to drive the interpretation of the Brenr record over the glacial-interglacial timescale.
To better stress this point and make clear that we proceeded in this direction knowing the discussed uncertainties on the deglaciation, we added the following sentence at the end of section 4.1:
*"Despite the application of the NGRIP temperature alone appears unable to fully represent the suggested two regime modelization during the deglaciation, possibly due to the particular conditions proper of this period, we suggest that at glacial-interglacial timescales such a temperature-based discrimination could be still used to represent the two regime variability. "*

- We have not shown for clarity of the figure the experimental errors propagated from in the calculation of the $B_{renr,Mg}$ values, but, being in the order of ±1, the points mentioned by the Reviewer (12.5, 13.8, but also the 11 kyr) are within 1σ of each other. Thus, we believe that any more-detailed interpretation would be highly speculative.

One minor style comment: normally, equations are produced at their first mention in the text. This is not the case for equations 1 and 7. I am not sure this can be easily fixed without altering the flow of the text, but please give it a thought.
We agree that Eq. 1 is the final one, but conceptually the Method section is structured so that we first present how the climate-relevant quantity ($B_{renr}$, ssNa) should be calculated followed by the details, caveats and all different cases needed to perform such calculation. After some consideration we think that the other way round would be conceptually less clear to the readers.

In summary, it is inevitable that this paper will leave some unanswered questions, and the present work seems to me as worthy of publication in CP. I however recommend that the authors think of addressing the points I raised when submitting a final version.

[revised manuscript text omitted]
. In general, bromine-reactive surfaces would increase the atmospheric residence time of bromine species, enhancing the bromine-to-sodium ratios away from the sea ice or ocean sources. On this topic,  from springtime Arctic snow samples collected from a transect directed inland, Simpson et al. (2005) showed that sodium is  deposited faster than bromine, suggesting the role of longer lasting gas-phase bromine

 (Fig. 3 in their study). From samples collected during a coast-to-inland Antarctic transect (Zhongshan Station–Dome A), Li et al. (2014) showed much reduced spatial gradients between sodium and bromine snow concentrations, if compared to Simpson and co-workers (Fig. 2 in their study). In a similar inland-directed Antarctic transect (Talos Dome–Dome C), Spolaor et al. (2013b) presented simultaneous bromine and sodium deposition fluxes from 3 sites and a model run aimed at explaining the experimental results (Fig. 6 in their study). The authors note that for their results to be explained by the model, the deposition velocity of HBr needs to be set at least 3 times larger than the average sea-salt aerosol deposition velocity. To conclude, the interference of transport processes with the sea ice source effects in either the Arctic or Antarctica is not clear and should be further investigated.

[revised manuscript text omitted]

$$ssNa = \frac{Na - (Na/Mg)_t \cdot Mg}{1 - (Mg/Na)_m \cdot (Na/Mg)_t} \tag{7}$$

$$ssMg = \frac{(Mg/Na)_m \cdot Na - (Mg/Na)_m \cdot (Na/Mg)_t \cdot Mg}{1 - (Mg/Na)_m \cdot (Na/Mg)_t} \tag{8}$$

In the Holocene, on average ssNa≈0.99Na, while ssMg≈0.50-0.90Mg. During MIS2, ssNa≈0.6–0.7Na and ssMg≈0.20–0.30Mg, while during MIS4 ssNa≈0.50–0.60Na and ssMg≈0.10–0.20Mg. In those Holocene samples where magnesium was not measured, we assume Na = ssNa, making on average a 1% error.

The same procedure using calcium  instead of magnesium has been applied to calculate ssNa concentrations from Antarctic ice cores (Röthlisberger et al., 2002), by solving the equations similar to Eq. 7 and Eq. 8 with Mg replaced by Ca. In the RECAP record, such correction  using with Asian dust $(Na/Ca)_t=0.38$ and marine $(Ca/Na)_m = 0.038$ (Millero et al., 2008) unrealistically predicts ssNa ≈ 0 throughout the glacial period. Small changes in the $(Na/Ca)_t$ value do not significantly change this result. It appears that higher-than expected calcium is deposited at Renland, and therefore a lower $(Na/Ca)_t$ is to be used. Excess of calcium during the glacial period, has been reported in Greenlandic ice cores, in the form of calcite ($CaCO_3$), dolomite ($CaMg(CO_3)_2$) and gypsum ($CaSO_4 \cdot 2H_2O$) (Mayewski et al., 1994; Maggi, 1997; De Angelis et al., 1997). In their investigation of the GRIP core Legrand and Mayewski 1997 find larger increase in  sulfate ($SO_4^{2-}$)

in Greenland compared to Antarctica during glacial times. Such large enhancements of the sulfate level are well correlated with calcium increases, but not with MSA, suggesting that sulfate levels are related to nonbiogenic sulfur sources (gypsum emissions from deserts, for instance) (Legrand and Mayewski, 1997). Different calcium sources were likely active during the glacial as compared to warmer periods, but it is unclear whether these were new active continental sources or continental shelfs
5 which became exposed due to a lowered sea level. Similarly to what has been observed in other Greenland ice cores, extra calcium is also found in the RECAP core during the glacial, as illustrated from the scatter plot with magnesium (Fig. 3). From  this calcium-magnesium relation, we estimate that during the coldest and most arid glacial times $(Ca/Mg)_t = 23\pm1$ (blue fit in Fig. 3), while in other periods we assume that the composition remained the same as the modern Asian dust composition (Appendix B):

$$\left(\frac{Ca}{Mg}\right)_{t,modern} = 3.7 \tag{9}$$

This hypothesis is substantiated by the value found for the lower envelope for the Holocene, interstadial late MIS5 samples (green points in Fig. 3) as well as for composition of the bottom 22 meters of the core (562–584 m, red points in Fig. 3): $(Ca/Mg)_t = 3\pm1$ (red fit). We note that these bottom measurements do not appear in any time series since the core chronology ends at 120 kyr (562 m). We here suggest that the RECAP bottom 22 meters  date back to the previous
15 interglacial, the Eemian.

Sea-salt sodium concentrations (ssNa) can therefore be calculated using calcium, by replacing in Eq. 7 Mg→Ca and by using $(Ca/Na)_m = 0.038$ and $(Na/Ca)_t$ calculated by:

$$\left(\frac{Na}{Ca}\right)_t = \frac{(Na/Mg)_{t,modern}}{(Ca/Mg)_{t,glacial\,fit}} = \frac{1.23}{23} = 0.054 \tag{10}$$

 As a comparison,
20 in the GRIP core, where the ssNa calculations were based on a mixed chlorine-calcium method, De Angelis et al. 1997 found empirically that $(Na/Ca)_{t,glacial} = 0.036$ in glacial ice (significant correlation, $\rho=0.7$) and $(Na/Ca)_{t,modern} = 0.07$ (weak correlation, $\rho=0.2$) in Holocene ice. The calculation of RECAP nssCa reveals that calcium contains almost a purely crustal signature: $nssCa \gtrsim 0.95Ca$ throughout the record.

The ssNa curves calculated with magnesium and calcium ($ssNa_{Mg}$, $ssNa_{Ca}$) differ by less than $1\sigma$, while appreciable dif-
25 ferences between these two curves and the chlorine one ($ssNa_{Cl}$) are only found during MIS2 and MIS4 (Fig. 4b). A weak positive correlation ($\rho=0.32$, $p<10^4$, $n=144$) is found between Br and $ssNa_{Mg}$ concentrations in glacial ice (Appendix C, Fig. C1) suggesting that although the two elements share a common marine source, other source/transport effects are playing a role in driving changes in their respective ice concentrations.

[revised manuscript text omitted]
. Despite the application of the NGRIP temperature alone appears unable to fully represent the suggested two regime modelization during the deglaciation, possibly due to the particular conditions proper of this period, we suggest that at glacial-interglacial timescales such a temperature-based discrimination could be still used to represent the two regime variability.

**4.2 The 120,000 year $Br_{enr}$ record**

We now apply the previously mentioned temperature-based discrimination of the two sea ice regimes to the 120,000 year $Br_{enr,Mg}$ record (Fig. 7). The  regime type in each ice sample is represented by the color of the error bars: blue for the FYSI/MYSI regime and red for the FYSI/OW regime. In order to test the sensitivity of the regime output on the threshold value, 4 scenarios are computed, using a $\pm1\sigma$ and a $\pm2\sigma$ value around the temperature threshold mean value $\overline{T}$ = -44.6 °C (Fig. D1). Except for 2(1) samples at ca. 20 kyr showing a different regime type in the $\overline{T}$-2(1)$\sigma$  scenario during the deglaciation (Fig. D1), the regime discrimination mentioned in the following discussion is invariant with respect to the 4  temperature-based scenarios. The same analysis is also performed on the $Br_{enr,Cl}$ curves (Fig. D2).

MIS 5 is characterized by increasing $Br_{enr,Mg}$ values in the FYSI/OW regime from 120 to 80 kyr ago (Greenland Interstadial, GI-21, Fig. 7). We interpret this trend with increasing FYSI extent in the North Atlantic. From the end of GI-21 the $Br_{enr}$ regime changes to FYSI/MYSI as the NGRIP temperature drops during late-MIS 5 and MIS 4. As compared to the levels reached during GI-21 (8±1), lower $Br_{enr,Mg}$ values are found during GS-21, GS-20 and GS-19 (5±2) and on average during MIS 4 (7±1). We interpret the combined effect of decreased $Br_{enr,Mg}$ values and a FYSI/MYSI regime with an increase of MYSI extent in the North Atlantic from late-MIS 5 to MIS 4. The $Br_{enr,Cl}$ curve shows even lower values during MIS 4, predicting more extended MYSI than the $Br_{enr,Mg}$ record (Fig. D2). Overall, from 120 kyr ago to MIS 4, the change of $Br_{enr}$ regime (from FYSI/OW to FYSI/MYSI) is opposite to what is observed during the deglaciation (from FYSI/MYSI to FYSI/OW): from a relatively warm climate 120 kyr ago, the cooling trend is characterized by an increasing FYSI extent until the point (GI-21) in which the FYSI starts to be replaced by MYSI. This maximum $Br_{enr,Mg}$ value at the time of the regime change (GI-21) is similar to the peak $Br_{enr,Mg}$ value reached during the deglaciation: $Br_{enr,Mg}(12 \text{ kyr}) \simeq Br_{enr,Mg}(\text{GI-21}) \simeq 8$. The same result holds if one considers the $Br_{enr,Cl}$ serie (Fig. D2). Unlike the longer lasting GI-21, the Greenland Interstadials GI-20 and GI-19.2 display very low $Br_{enr}$ values ($\simeq$ 3-5). A possible explanation for these low values could be found in a fast replacement (at least not captured by the time resolution of the $Br_{enr}$ record) of MYSI by OW conditions. Higher time resolved measurements would be needed to test this hypothesis.

Moving from MIS 4 to MIS 3, the $Br_{enr,Mg}$ values remain within 1$\sigma$ of each other. This suggests similar FYSI extent in the two periods, although the FYSI/MYSI regime operating during MIS 4 could suggest extra MYSI extent during this period, as compared to MIS 3, the latter being generally characterized by a mixture of both $Br_{enr}$ regimes. This hypothesis is supported

by the $Br_{enr,Cl}$ series, which shows lower values during MIS 4 than MIS 3 (Fig. D2). A sea ice record in the Norwegian Sea (Hoff et al., 2016) also indicates more perennial sea ice conditions during MIS 4 than during MIS 3. The time resolution does not allow a deep investigation on DO events during MIS 3, although a low $Br_{enr}$ value in the FYSI/OW regime during the GI-12 ($Br_{enr,Mg}$(46.0-46.6 kyr)=5.2±0.7) suggests a shift to open water conditions during the interstadial, similarly to GI-19.2 and GI-20. Possibly, similar sea ice dynamics were in play during these stadial-interstadial transitions. Increased time resolution is however needed to better characterize  DO events.

MIS 2 is characterized by a FYSI/MYSI regime and progressively decreasing  $Br_{enr,Mg}$  values (Fig.  7e), with a minimum reached ~23 kyr ago, during the Last Glacial Maximum: $Br_{enr,Mg}$(LGM)=3.3±0.3. We interpret this negative $Br_{enr,Mg}$ (and $Br_{enr,Cl}$, Fig. D2) trend with progressively increasing MYSI conditions in the whole North Atlantic, that reached a maximum during the LGM.  Relatively high and increasing PIP$_{25}$ values  (≈0.5-1) are found in the Norwegian Sea  record (Fig. 7d), with maximum sea ice  registered ca. 20-23 kyr ago. In broad agreement with the Norwegian record is the sea ice record from Fram Strait Fig. (7c), the latter showing high (≈0.5-1) and variable PIP$_{25}$ values until the longer lasting 19 kyr maximum. Overall, the cold MIS2 period is characterized by decreasing seasonal sea ice and increasing multiyear sea ice with a maximum reached ca. 20 kyr ago.

[revised manuscript text omitted]

Taranczewski, T., Freitag, J., Eisen, O., Vinther, B., Wahl, S., and Kipfstuhl, S.: 10,000 years of melt history of the 2015 Renland ice core, EastGreenland, The Cryosphere Discussions, 2019, 1–16, 
[revised manuscript text omitted]
)  $PIP_{25}$ record from the  Svalbard margin (, core MSM5/5-712-2 from Müller and Stein 2014 ) d) 90 kyr $P_DIP_{25}$ and $P_BIP_{25}$ records from the Norwegian Sea presented as 5-point running mean(from Hoff et al. 2016). ) discrimination of the $Br_{enr,Mg}$ regimes computed according to the integrated temperature value with respect to the threshold. The FYSI/OW

**Appendix A: Bromine loss from the snowpack**

The stability of bromine after deposition has been questioned since the first investigation carried out by Dibb et al. (2010) in Summit (Greenland). We summarize here the experimental investigations that have been carried out on this topic ever since both in Antarctica and in the Arctic.

**A1 Antarctica**

McConnell et al. (2017) investigated simultaneous sodium and bromine fluxes across Antarctica (see their figure S4 in the appendix showing the reconstructed fluxes from an array of 11 cores measured in McConnell et al. (2014)). In particular, the J(Na) and J(Br) - Accumulation rate (A) linear relations (impressive J(Br)-A linear fit from 50 to 400 kg m$^{-2}$ yr$^{-1}$) suggest that, while sodium loss is negligible, bromine reemission is quantified to $17\pm2$ $\mu$g m$^{-2}$ yr$^{-1}$ (negative intercept, i.e. snow-air flux). According to the results, the bromine loss from Antarctic snowpack would be (note that this exercise works on the hypothesis the bromine air concentration is the same above all sites, but the goodness of their fit would suggest that this is the case): 65% at sites with A=50 kg m$^{-2}$ yr$^{-1}$, 32% at sites with A=100 kg m$^{-2}$ yr$^{-1}$, 22% at sites with A=150 kg m$^{-2}$ yr$^{-1}$, 16% at sites with A=200 kg m$^{-2}$ yr$^{-1}$, 13% at sites with A=250 kg m$^{-2}$ yr$^{-1}$, 11% at sites with A=300 kg m$^{-2}$ yr$^{-1}$. These negative fluxes would be representative of Antarctic conditions for the duration of the records, i.e. few centuries to max 2 millennia (see Supp. Table 2 in the 2014 ref). In their Dome C (A=30 kg m$^{-2}$ yr$^{-1}$) study, Legrand et al. (2016) analyzed bromine content in a 110cm 2-cm resolution snowpit. Their measurements show that the upper 2 cm contain 1.25 times more bromine that the 2-4 cm layer, 2.5 times more than the 4-8 cm layer and ca. 10 times more than the 8-12 cm layer. Based on the higher bromine concentrations towards the surface the authors suggest a possible bromine remobilization from the snow. In testing the hypothesis, the authors model whether the measured bromine snow content is high enough to sustain the 1.7 pptv air concentration of Br$_y$ they measured in the overlying atmosphere. They find that to explain such value, the snow bromine storage should be 35 times larger than the actual measurements. They therefore conclude that in Dome C their investigation do not support the importance of snowpack bromine emissions. This conclusion contrasts with what would be concluded by McConnell and co-workers.

From a surface snow experiments carried out in Dome C in 2014 (Antarctica, A=30 kg m$^{-2}$ yr$^{-1}$) by Spolaor et al. (2018) , the authors observe constant bromine concentrations from mid December to mid January (austral summer). In a similar experiment carried out the following year, from late November to late December 2015, higher bromine concentrations were found in November (late spring) with a decreasing trend towards the end of the year. The authors relate such drop to a change in the airmass transport, coastal followed by inland pathways, as inferred from back trajectory modeling, rather than to bromine loss from the snow.

From another surface snow experiment carried out at GV7 (Antarctica, A=270 kg m$^{-2}$ yr$^{-1}$), Vallelonga et al. (QSR, in review) compare bromine fresh surface snow concentrations with snowpit values, and suggest, on the basis of similar values between the values in the surface samples and in the surface of the snowpit (that would integrate the values of the surface samples), that at this site bromine is stable after deposition (McConnell calculation would predict a loss of 12%). Looking at

the 100-300 $\mu$g m$^{-2}$ yr$^{-1}$ bromine annual (2010, 2011, 2012, 2013) fluxes calculated along the Talos Dome – GV7 traverse (Maffezzoli et al., 2017), a negative flux of 17$\pm$2 $\mu$g m$^{-2}$ yr$^{-1}$ would imply that, if present, the bromine loss can be quantified in this region as 6-17%. As McConnell and co-workers point out, it is surprising that the snow-air bromine flux they found is not dependent on snow accumulation (their fit appears linear until the last point, 400 kg m$^{-2}$ yr$^{-1}$).

**A2 Arctic**

In the Arctic, the most robust piece of evidence result from hourly resolved surface snow measurements carried out in coastal Svalbard, showing that bromine photolytic loss cannot be appreciated across night-day cycles (Spolaor et al. 2019, see Fig. 6).

In an experiment described in Vallelonga and coworkers (QSR, in review) carried out in Ny-Ålesund (Svalbard, A=600 kg m$^{-2}$ yr$^{-1}$), sodium and bromine were measured in surface snow daily from April to June (spring to summer). From April to late May (afterwards surface melting and positive temperatures led to 80% sodium drainage from the snowpack) Br$_{enr}$ values decreased by a factor 2 (max). It is difficult however, to distinguish whether this decrease is due to bromine reemission or to early spring bromine explosions from sea ice surfaces, located ca. 150 km away from the sampling site, enriching the spring deposited snow layers.

**A3 Conclusive remarks on bromine loss**

To conclude, field experiments aimed at quantifying bromine loss from the snowpack are made difficult by the concurrent effect of other variables (source effects, changes in transport pathways, other post depositional processes) that can act simultaneously. Estimates of bromine loss from Dome C (Antarctica) have led to not unique conclusions. This site is likely one of the most challenging to carry out this type of study since the snow accumulation is so limited that it is challenging to decipher whether any loss can be attributed to reemission or wind erosion processes, exposing older layers having generally different bromine concentrations. In other parts of Antarctica, McConnell et al. (2017) suggest bromine snow-air flux of loss of 17$\pm$2 $\mu$g m$^{-2}$ yr$^{-1}$ regardless of snow accumulation, while at GV7 bromine loss was not detected (Vallelonga et al., QSR, in review). In the Arctic, photolytic loss was not detected in Svalbard. Other experiments have been shown inconclusive due to the presence of surface melting. At sites like Renland (where Holocene melting occurs, see Taranczewski et al. 2019), the extremely high snow accumulation could in general suggest reduced bromine loss, but again, the fivefold decrease snow accumulation during the glacial (preliminary A$_{glacial,stadials}$ = 100 kg m$^{-2}$ yr$^{-1}$ (Vinther unpublished)) may introduce a possible bromine loss during periods of reduced accumulation, thus increasing Br$_{enr}$ values during the coldest parts of the record. To test these hypotheses and especially the bromine loss during the glacial, a surface study should be carried out at a 100 kg m$^{-2}$ yr$^{-1}$ accumulation site in Greenland. To conclude, at present there is no enough data available to quantify the possible bromine loss from the Renland ice cap but, if present and dependent on accumulation, the effect would be to increase the measured Br$_{enr}$ values during the coldest sections of the glacial period.

[revised manuscript text omitted]